# Human DCP1 is crucial for mRNA decapping and possesses paralog-specific gene regulating functions

**Ting-Wen Chen[1,2,3†], Hsiao-Wei Liao[4†], Michelle Noble[5], Jing-Yi Siao[6], Yu-Hsuan Cheng[6], Wei-Chung Chiang[6], Yi-Tzu Lo[1], Chung-Te Chang[5,6*]**

[1]Institute of Bioinformatics and Systems Biology, National Yang Ming Chiao Tung University, Hsinchu, Taiwan; [2]Department of Biological Science and Technology, National Yang Ming Chiao Tung University, Hsinchu, Taiwan; [3]Center for Intelligent Drug Systems and Smart Bio-devices (IDS2 B), National Yang Ming Chiao Tung University, Hsinchu, Taiwan; [4]Department of Pharmacy, National Yang Ming Chiao Tung University, Taipei City, Taiwan; [5]Department of Biochemistry, Max Planck Institute for Developmental Biology, Tübingen, Germany; [6]Institute of Biochemistry and Molecular Biology, National Yang Ming Chiao Tung University, Taipei, Taiwan

**\*For correspondence:**
chungte.chang@gmail.com

[†]These authors contributed equally to this work

**Competing interest:** The authors declare that no competing interests exist.

## eLife Assessment

This study attempts to understand the functional roles of the human DCP1 paralogs in regulating RNA decay by DCP2. Using a combination of cellular-based assays and in vitro assays, the authors conclude that DCP1a/b plays a role in regulating DCP2 activity. While this revised version presents some new and interesting observations on human DCP1, the underlying data to support its claims remain **incomplete**. Overall, these results will be **useful** to the RNA community.

**Abstract** The mRNA 5'-cap structure removal by the decapping enzyme DCP2 is a critical step in gene regulation. While DCP2 is the catalytic subunit in the decapping complex, its activity is strongly enhanced by multiple factors, particularly DCP1, which is the major activator in yeast. However, the precise role of DCP1 in metazoans has yet to be fully elucidated. Moreover, in humans, the specific biological functions of the two DCP1 paralogs, DCP1a and DCP1b, remain largely unknown. To investigate the role of human DCP1, we generated cell lines that were deficient in DCP1a, DCP1b, or both to evaluate the importance of DCP1 in the decapping machinery. Our results highlight the importance of human DCP1 in decapping process and show that the EVH1 domain of DCP1 enhances the mRNA-binding affinity of DCP2. Transcriptome and metabolome analyses outline the distinct functions of DCP1a and DCP1b in human cells, regulating specific endogenous mRNA targets and biological processes. Overall, our findings provide insights into the molecular mechanism of human DCP1 in mRNA decapping and shed light on the distinct functions of its paralogs.

## Introduction

Gene expression must be stringently regulated to ensure efficient control of cellular processes. A key mechanism of gene expression regulation is the control of mRNA degradation (*Chen and Shyu, 2017*; *Corbett, 2018*), mediated by the removal of protective structures from mRNA, which suppresses translation or leads to mRNA degradation. In most eukaryotic cells, mRNA is stabilized through the addition during post-transcriptional modification of an $N^7$-methyl guanosine (m[7]G) cap at the 5' end

and a polyA tail at the 3′ end (*Furuichi et al., 1977*; *Garneau et al., 2007*). General mRNA decay is typically initiated by deadenylation, which is mediated by the coordinated activity of the CCR4/NOT and PAN2/3 complexes (*Chen and Shyu, 2011*; *Decker and Parker, 1993*). Shortening of the polyA tail typically initiates mRNA decay, resulting in exonucleolytic degradation in the 5′–3′ or 3′–5′ direction, respectively, facilitated by the exonuclease XRN1 or the exosome complex (*Chang et al., 2019*; *Garneau et al., 2007*; *Lim et al., 2016*; *Subtelny et al., 2014*; *Webster et al., 2018*; *Yi et al., 2018*). While the cytoplasmic RNA exosome degrades mRNA in the 3′–5′ direction, the primary effect of deadenylation is 'mRNA decapping', the removal of the m⁷G cap structure at the 5′-end of the mRNA that otherwise prevents mRNA degradation (*Stevens, 1988*). This connection between deadenylation and decapping is facilitated by Pat1, which, together with the Lsm1-7 complex, recruits the decapping complex through a combination of disordered and structured domains, promoting distinct steps of mRNA decay (*Charenton et al., 2017*; *Charenton and Graille, 2018*; *Fourati et al., 2014*; *Lobel and Gross, 2020*; *Lobel et al., 2019*; *Sharif and Conti, 2013*; *Wu et al., 2014*).

The mRNA decapping process is crucial for gene regulation because it leads to the irreversible removal of the mRNA cap structure, inhibiting translation initiation and inducing the XRN1-mediated degradation of mRNA through the 5′–3′ mRNA decay pathway (*Hsu and Stevens, 1993*; *Topisirovic et al., 2011*). Decapping does not only lead to the degradation of bulk mRNA molecules; it is also involved in specific mRNA decay pathways, such as AU-rich element-mediated decay, nonsense-mediated decay (NMD), and miRNA-triggered mRNA turnover (*Arribas-Layton et al., 2013*; *Chen and Shyu, 2003*; *Fenger-Grøn et al., 2005*; *Jonas and Izaurralde, 2015*; *Loh et al., 2013*). Because of its crucial role in gene regulation, decapping is tightly regulated by the activity of both general and pathway-specific factors through various mechanisms. In the context of NMD for instance, target transcripts can be degraded through endonucleolytic cleavage by SMG6 (*Eberle et al., 2009*; *Huntzinger et al., 2008*), or be facilitated by SMG5 in conjunction with the NMD effector SMG7 or the decapping factor PNRC2, leading to enhanced deadenylation and decapping (*Cho et al., 2009*; *Fukuhara et al., 2005*; *Loh et al., 2013*). Understanding the determinants of the preferential associations between certain degradation mechanisms and particular transcripts requires further study (*Metze et al., 2013*). Overall, the pivotal role of decapping in the intricate network of mRNA decay justifies ongoing research in this field.

In eukaryotes, the removal of the mRNA 5′-m⁷GDP cap structure is catalyzed by the decapping enzyme DCP2, a pyrophosphatase belonging to the Nudix family of proteins. This process results in the generation of 5′ monophosphorylated mRNA, which is subsequently targeted for degradation (*Arribas-Layton et al., 2013*; *Lykke-Andersen, 2002*; *Steiger et al., 2003*; *van Dijk et al., 2002*; *Wang et al., 2002*). The central component of the complex, DCP2, has two major domains: the N-terminal regulatory domain (NRD) and the catalytic Nudix domain (*She et al., 2006*; *She et al., 2008*). The positively charged patch of the Nudix domain plays a crucial role in RNA binding (*Deshmukh et al., 2008*; *Piccirillo et al., 2003*). Meanwhile, the NRD enhances decapping activity by specifically recognizing the m⁷G nucleotide of the mRNA cap structure and interacts with the N-terminal EVH1 domain of the major activator DCP1, further accelerating the decapping process (*Chang et al., 2014*; *Charenton et al., 2016*; *Dunckley and Parker, 1999*; *Floor et al., 2010*; *Mugridge et al., 2016*; *She et al., 2008*; *Wang et al., 2002*; *Wurm et al., 2017*). Yeast Dcp2 has an additional unstructured C-terminal extension that contains short helical leucine-rich motifs (HLMs) responsible for protein–protein interactions and elements that inhibit Dcp2 catalytic activity (*Charenton et al., 2017*; *He and Jacobson, 2015*). Interestingly, in contrast to yeast, metazoans have the HLM in DCP1, implying a conserved biological function with evolving regulation of mRNA decapping (*Fromm et al., 2012*).

The interactions between DCP1 and DCP2 proteins and their functions across various species have garnered extensive attention in the study of mRNA degradation mechanisms. While DCP2 is capable of demonstrating catalytic activity on its own (*Lykke-Andersen, 2002*; *van Dijk et al., 2002*; *Wang et al., 2002*), it is important to note that additional cofactors are involved in stimulating DCP2 for complete activation. In yeast, the primary activator of DCP2 is Dcp1, which is essential for decapping activity in vivo (*Beelman et al., 1996*) and considerably enhances the catalytic activity of Dcp2 in vitro (*Deshmukh et al., 2008*; *Floor et al., 2010*; *She et al., 2006*). Moreover, DCP1 has been shown to interact with other decapping proteins, such as DCP2, to form functional decapping complexes critical for mRNA degradation. Localization and functional analysis of decapping proteins across various species indicate that Dcp1-dependent decapping mechanisms have been conserved throughout

evolution and play a significant role in the mRNA decapping machinery (*Lall et al., 2005*; *Sakuno et al., 2004*). Furthermore, the importance of DCP1 in plant species underscores the necessity of the decapping complex for proper plant development and highlights its potential homology with yeast and human DCP1 (*Xu et al., 2006*). These findings suggest that the interaction between DCP1 and DCP2 and their role in mRNA degradation is a conserved key biological process across species.

In addition to Dcp1, proteins such as Edc1-3, Pby1, Lsm1-7, Pat1, and Dhh1 (PNRC1, PNRC2, EDC3, EDC4, Lsm1-7, Lsm14, PatL1, and DDX6 in metazoans) promote decapping by directly or indirectly interacting with Dcp2 (*Charenton et al., 2020*; *Charenton et al., 2017*; *Charenton and Graille, 2018*; *Charenton et al., 2016*; *Cho et al., 2009*; *Gaviraghi et al., 2018*; *He and Jacobson, 2023*; *He et al., 2022*; *Jonas and Izaurralde, 2013*; *Ling et al., 2011*; *Lobel et al., 2019*; *Mugridge et al., 2018b*). These proteins facilitate decapping through the assembly of messenger ribonucleoprotein (RNP) units or the structural rearrangement of DCP2. High levels of these proteins have been detected in large RNP granules, processing bodies (P-bodies), which likely serve as sites for the storage of translationally repressed RNAs and the process of mRNA decay (*Eulalio et al., 2007*; *Jonas and Izaurralde, 2013*; *Luo et al., 2018*).

The interactions between the subunits of the decapping complex are crucial for enhancing its activity and for the targeting of specific mRNAs. Although Dcp1 and Dcp2 have been shown to interact strongly in yeast, direct DCP1-DCP2 interactions are weak in metazoans. Instead, the metazoan-specific protein EDC4, which has binding sites for both DCP1 and DCP2, likely functions as a scaffold that brings these proteins together despite their weak interaction propensity (*Chang et al., 2014*; *Fenger-Grøn et al., 2005*). In addition to EDC4, DCP1 acts as a binding platform for other coactivator proteins, such as yeast Edc1 and Edc2 and the human PNRC paralogs, PNRC1 and PNRC2 (*Gaviraghi et al., 2018*; *Jonas and Izaurralde, 2013*; *Mugridge et al., 2016*). In metazoans, the HLM domain of DCP1 interacts with the Lsm domains of proteins such as EDC3 and Lsm14 in a mutually exclusive manner. Moreover, in multicellular eukaryotes, DCP1 features a C-terminal extension that induces DCP1 trimerization, referred to as the trimerization domain (TD). Trimerization is a prerequisite for the incorporation of DCP1 into active decapping complexes and for promoting efficient mRNA decapping in vivo (*Fromm et al., 2012*; *Jonas and Izaurralde, 2013*; *Tritschler et al., 2007*). This competition results in the formation of structurally and functionally distinct assemblies, contributing to target specificity (*Vidya and Duchaine, 2022*).

Recent structural analyses of the *Kluyveromyces lactis* Dcp1-Dcp2-Edc3 complex indicate that the Lsm domain of Edc3 binds to an extended helix at the C-terminus of the Dcp2 Nudix domain (*Charenton et al., 2016*). Furthermore, the activation peptide of Edc1 bridges Dcp1 and Dcp2, as observed in the Edc1-Dcp1-Dcp2 complex in *Saccharomyces pombe* and *Kluyveromyces lactis* (*Mugridge et al., 2018b*; *Wurm et al., 2017*). These structural insights suggest that Edc1 and Edc3 collaborate through distinct mechanisms, aiding the decapping process (*Charenton and Graille, 2018*; *Charenton et al., 2016*; *Mugridge et al., 2018a*; *Mugridge et al., 2018b*; *Tibble et al., 2021*). Specifically, Edc1 stabilizes Dcp2's catalytically active conformation, while Edc3 extends the RNA binding surface of the Dcp2 Nudix domain (*Charenton et al., 2016*; *Mugridge et al., 2018b*). This extended RNA interaction potentially accounts for the transcript-specific nature of Edc3-mediated decapping activation, hinting at distinct coactivators' roles in gene-specific regulation.

While structural studies suggest that interactions with Dcp1 are central to the activation of Dcp2 (*Charenton and Graille, 2018*; *Deshmukh et al., 2008*; *Mugridge et al., 2018a*; *Mugridge et al., 2016*; *She et al., 2008*; *Valkov et al., 2016*), the lack of structural information on the human decapping complex hinders our understanding of the specific role of human DCP1 in decapping. It remains unclear how DCP1 contributes to the conformational changes in the DCP2 catalytic cycle or to mRNA and cap structure recognition. A further complication is the existence of two paralogs of human DCP1, DCP1a, and DCP1b, which have 31% sequence identity over their entire length (*Lykke-Andersen, 2002*; *van Dijk et al., 2002*). The functional differences of these paralogs have yet to be determined. Studies of their potentially distinct roles in the decapping complex may provide new insight into the molecular mechanisms of mRNA decapping. The fact that decapping dysfunctions can have detrimental effects on cell growth and have been linked to severe neurological disorders in humans (*Ahmed et al., 2015*; *Ng et al., 2015*) highlights the importance of understanding the mechanisms of mRNA decapping regulation and the potential therapeutic implications.

In this study, we investigated the roles of human DCP1 in mRNA decapping by generating HEK-293T cell lines lacking DCP1a, DCP1b, or both. Our findings indicate that human DCP1 promotes decapping in vivo and that DCP1a and DCP1b play redundant roles in the general 5'–3' mRNA decay pathway. We found that EVH1 domain may be crucial for enhancing DCP2's recruitment to target mRNA in cells. Multiomics analyses revealed distinct expressional and metabolomic profiles for the knockout cell lines. These results indicate that the DCP1 paralogs have distinct roles in the regulation of various biological processes. Overall, our findings offer valuable insights into the mechanisms underlying human DCP1-mediated mRNA decapping and the specific mRNA targets and biological processes regulated by DCP1a and DCP1b in humans.

## Results

### Knockout of human DCP1 represses the general mRNA decapping process in vivo

To investigate the roles of the two DCP1 paralogs, we knocked out DCP1a, DCP1b, or both in HEK-293T cells using the CRISPR-Cas9 technique. Western blot analysis and Sanger sequencing confirmed the successful generation of the knockout cell lines (*Figure 1—figure supplement 1A–C*). We further investigated whether the absence of DCP1a, DCP1b, or both affected decapping. To monitor mRNA decay, the NMD factor SMG7 was tethered to an mRNA reporter in the DCP1a/b-knockout cells (*Figure 1A*). While DCP1 is the main decapping activator in yeast, previous studies in metazoans have shown that at least two decapping activators have to be simultaneously depleted to effectively inhibit mRNA decapping (*Braun et al., 2012*; *Sakuno et al., 2004*). Surprisingly, we observed that the depletion of both DCP1a and DCP1b substantially inhibited SMG7-mediated mRNA degradation, resulting in the accumulation of deadenylated reporter RNA molecules (*Figure 1B*, lane 2 and 6).

To investigate whether the inhibitory effect of DCP1 knockout on decapping is specific to SMG7 or a general effect, we performed tethering assays using either the mRNA decay factor Nanos1 or the crucial miRNA component TNRC6A to induce 5'–3' mRNA decay. In both cases, DCP1 knockout led to the substantial accumulation of mRNA decay intermediates (*Figure 1—figure supplement 2A–C*), indicating that DCP1 is important for the decapping and degradation of mRNA in general.

To confirm the presence of a cap structure on the accumulated reporter RNA intermediate, we treated it with Terminator 5'-phosphate exonuclease. The resistance of the transcripts to this treatment is strong evidence that the accumulated RNA fragments had intact 5' caps (*Figure 1—figure supplement 2D*). The accumulation of capped mRNA decay intermediates suggests slowed or impaired DCP2-mediated decapping in the absence of DCP1. Taken together, these findings provide compelling evidence for the important role of DCP1 in the decapping process in human cells, mirroring its established role in yeast.

To better understand the functional roles of the DCP1 paralogs in decapping, we performed tethering assays and reintroduced GFP-tagged DCP1a or DCP1b into DCP1a/b-knockout cells, leading in both cases to the restoration of decapping and degradation of the reporter mRNA (*Figure 1C and D*, *Figure 1—figure supplement 2E*). This finding strongly suggests that DCP1a and DCP1b are functionally redundant for the general decapping of mRNA.

### DCP1 EVH1 domain is crucial for cellular mRNA decapping process

Building on our finding that DCP1 is crucial for efficient mRNA decapping in human cells, we sought to identify the specific region of DCP1 that is important for the function of the decapping complex. Given the observed functional redundancy of DCP1a and DCP1b in the decapping process, we chose to use DCP1a as a model (*Figure 1E*). In DCP1a/b knockout cells, we overexpressed DCP1a fragments lacking either of three major domains (EVH1, HLM, TD) and monitored mRNA degradation status (*Figure 1F–H*). Strikingly, we observed that overexpression of a DCP1a fragment lacking EVH1 domain (ΔEVH1) could not rescue the decapping defect, indicating that this domain is crucial to coordinate the cellular decapping process. Overexpression of DCP1a without the HLM domain (ΔHLM) partially restored the decapping defect, indicating a cooperative role for this domain in coordinating DCP1's decapping function. Adequate mRNA degradation occurred in cells overexpressing DCP1a fragment without TD domain.

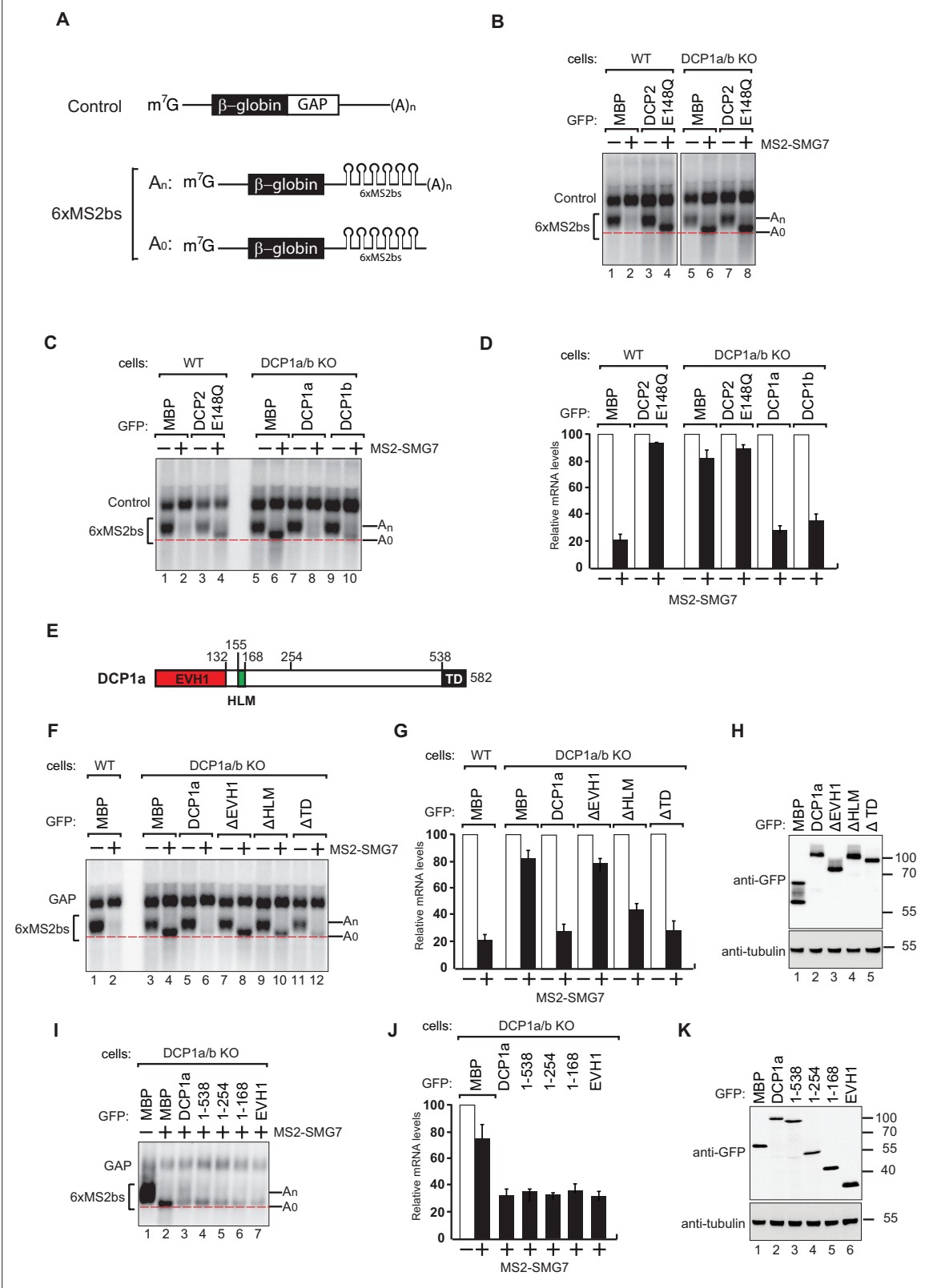

**Figure 1.** DCP1 is essential for human decapping process. (**A**) Schematic of the reporters used in tethering assays. (**B**) Wild-type or DCP1a/b knockout HEK-293T cells were transfected with a mixture of plasmids. One plasmid expressed the β-globin-6xMS2bs, and another plasmid expressed the transfection control, which contained the β-globin gene fused to the GAPDH 3′ UTR, but it lacked the MS2 binding sites (β-globin-GAP, Control). The third plasmid expressed MS2-HA or MS2-tagged SMG7 proteins, and a fourth plasmid encoding a GFP-tagged protein was included in the transfection

*Figure 1 continued on next page*

*Figure 1 continued*

mixtures where indicated. A northern blot of representative RNA samples is shown. The positions of the polyadenylated (An) and the deadenylated (A0) forms of the β-globin-6xMS2bs reporters are indicated on the right. A red dotted line additionally marks the fast migrating deadenylated (A0) form. The DCP2 inactive mutant (E148Q) serves as a negative control, highlighting the importance of DCP1 in decapping by showing the absence of decapping activity without functional DCP2. All experimental results were independently repeated at least three times. (**C**) Complementation assays with GFP-DCP1a or GFP-DCP1b constructs in HEK-293T DCP1a/b-null cells were performed similarly to panel (**A**). All experimental results were independently repeated at least three times. This experiment demonstrates whether the reintroduction of DCP1a or DCP1b can restore the decapping activity in cells lacking both DCP1 isoforms. (**D**) The β-globin-6xMS2bs mRNA levels were normalized to those of the control mRNA. These normalized values were set to 100 in cells expressing MS2-HA (white bars). The mean values for relative mRNA levels in cells expressing MS2-SMG7 were estimated with standard deviations (SD) from three independent experiments (black bars). This quantification provides a comparative measure of the mRNA stability under different conditions. (**E**) The domain organization of human DCP1a. (**F**) Complementation assays with GFP-DCP1a deletion constructs in HEK-293T DCP1a/b-null cells performed similarly to panel (**A**). All experimental results were independently repeated at least three times. This experiment helps identify which domains of DCP1a are crucial for its function in the decapping process. (**G**) The β-globin-6xMS2bs mRNA levels were normalized to those of the control mRNA. These normalized values were set to 100 in cells expressing MS2-HA (white bars). The mean values for relative mRNA levels in cells expressing MS2-SMG7 were estimated with SDs from three independent experiments (black bars). This data provides insight into the functional importance of different DCP1a domains. (**H**) A western blot demonstrating equivalent expression of the GFP-tagged proteins in panel (**F**). Tubulin served as a loading control, confirming that the expression levels of the deletion constructs were comparable. (**I**) Complementation assays with GFP-DCP1a fragment constructs in HEK-293T DCP1a/b-null cells performed similarly to panel (**A**). All experimental results were independently repeated at least three times. This set of experiments further delineates the specific regions of DCP1a necessary for its decapping activity. (**J**) The β-globin-6xMS2bs mRNA levels were normalized to those of the control mRNA. These normalized values were set to 100 in cells expressing MS2-HA (white bars). The mean values for relative mRNA levels in cells expressing MS2-SMG7 were estimated with SDs from three independent experiments (black bars). This helps validate the findings regarding the essential regions of DCP1a. (**K**) A western blot demonstrating equivalent expression of the GFP-tagged proteins in panel (**I**). Tubulin served as a loading control, ensuring that the variations in mRNA levels were due to functional differences in the DCP1a fragments rather than differences in protein expression.

The online version of this article includes the following source data and figure supplement(s) for figure 1:

**Source data 1.** Original file for the northern blot analysis in *Figure 1B*.

**Source data 2.** File containing *Figure 1B* and original scans of the relevant northern blot analysis with highlighted bands and sample labels.

**Source data 3.** Original file for the northern blot analysis in *Figure 1C*.

**Source data 4.** File containing *Figure 1C* and original scans of the relevant northern blot analysis with highlighted bands and sample labels.

**Source data 5.** Original file for the northern blot analysis in *Figure 1F*.

**Source data 6.** File containing *Figure 1F* and original scans of the relevant northern blot analysis with highlighted bands and sample labels.

**Source data 7.** Original file for the western blot in *Figure 1H*.

**Source data 8.** File containing *Figure 1H* and original scans of the relevant western blot analysis with highlighted bands and sample labels.

**Source data 9.** Original file for the northern blot analysis in *Figure 1I*.

**Source data 10.** File containing *Figure 1I* and original scans of the relevant northern blot analysis with highlighted bands and sample labels.

**Source data 11.** Original file for the western blot in *Figure 1K*.

**Source data 12.** File containing *Figure 1K* and original scans of the relevant western blot analysis with highlighted bands and sample labels.

**Figure supplement 1.** Verification of DCP1a and DCP1b knockout in HEK-293T cells.

**Figure supplement 1—source data 1.** Original file for the western blot analysis in *Figure 1—figure supplement 1A*.

**Figure supplement 1—source data 2.** *Figure 1—figure supplement 1A* and original scans of the relevant western blot analysis with highlighted bands and sample labels.

**Figure supplement 1—source data 3.** Original file for the western blot analysis in *Figure 1—figure supplement 1B*.

**Figure supplement 1—source data 4.** *Figure 1—figure supplement 1B* and original scans of the relevant Western blot analysis with highlighted bands and sample labels.

**Figure supplement 2.** Tethering assays and RNA stability analysis in DCP1a/b knockout cells.

**Figure supplement 2—source data 1.** Original file for the northern blot analysis in *Figure 1—figure supplement 2A*.

**Figure supplement 2—source data 2.** File containing *Figure 1—figure supplement 2A* and original scans of the relevant northern blot analysis with highlighted bands and sample labels.

**Figure supplement 2—source data 3.** Original file for the western blot analysis in *Figure 1—figure supplement 2C*.

**Figure supplement 2—source data 4.** *Figure 1—figure supplement 2C* and original scans of the relevant western blot analysis with highlighted bands and sample labels.

**Figure supplement 2—source data 5.** Original file for the northern blot analysis in *Figure 1—figure supplement 2D*.

*Figure 1 continued*

**Figure supplement 2—source data 6.** *Figure 1—figure supplement 2D* and original scans of the relevant northern blot analysis with highlighted bands and sample labels.

**Figure supplement 2—source data 7.** Original file for the western blot analysis in *Figure 1—figure supplement 2E*.

**Figure supplement 2—source data 8.** *Figure 1—figure supplement 2E* and original scans of the relevant western blot analysis with highlighted bands and sample labels.

To precisely identify the minimal functional region of DCP1, we generated a series of DCP1a expression constructs by sequentially removing its domains. The structured core of DCP1 comprises its N-terminal EVH1 and HLM domains, followed by a long unstructured region (amino acids 169–538) and the terminal trimeric domain (TD). Given the presence of this unstructured region, we utilized AlphaFold to predict areas potentially possessing secondary or tertiary structures. This approach enabled us to design more effective truncations. We began by referencing an AlphaFold prediction indicating that the DCP1a segment spanning amino acids 1–254 is structured (AlphaFoldDB: A0A087WT55). Using these predictions, we systematically deleted specific domains and regions. This stepwise truncation method allowed us to pinpoint the crucial segments required for DCP1a's function with greater accuracy. Overexpression and subsequent complementation assays in DCP1a/b-knockout cells revealed that all the constructs containing the EVH1 domain successfully restored the decapping ability of the enzyme complex (*Figure 1I–K*). Together, these findings indicate that the EVH1 domain is of DCP1 is essential for the formation of a functional decapping complex.

## DCP1 knockout does not affect the formation of P-bodies or the enzymatic activity of DCP2

Since we observed that DCP1 appears to be indispensable for mRNA decapping, we sought to identify the precise role of DCP1 in decapping. To begin with, we explored P-body formation and DCP2 localization in DCP1a/b-knockout cells. Immunofluorescence and confocal microscopy analysis of P-body formation indicated that DCP1 has little involvement in P-body formation. In DCP1a/b-knockout cells, DCP2 remained localized in P-body granules, indicating no correlation between DCP1 and P-body formation or DCP2 localization (*Figure 2A*, *Figure 2—figure supplement 1A and B*). Meanwhile, we observed that in DCP1a/b knockout cells, although the number of P-bodies only slightly increased, their size significantly enlarged (*Figure 2B*, *Figure 2—figure supplement 1C*). This suggests that the knockout of DCP1a/b, while not affecting P-body formation, still has some impact on the morphology of P-bodies.

Next, we wanted to clarify if the absence of cellular DCP1 has a direct effect on decapping activity of DCP2. To evaluate the effects of DCP1 on the decapping activity of DCP2, we performed in vitro decapping assays by immunoprecipitating DCP2 extracted from wild-type and DCP1a/b-knockout cells. We observed a decrease in the enzymatic activity of DCP2 immunopurified from DCP1a/b knockout cells, but it still retains a considerable decapping ability (*Figure 2C and D*, *Figure 2—figure supplement 1D*). These data suggest that, in accordance with prior literature, DCP2 is sufficient for in vitro decapping without DCP1 (*Lykke-Andersen, 2002*; *van Dijk et al., 2002*; *Wang et al., 2002*). Nevertheless, our previous observations emphasize the significant role of DCP1a/b gene knockout in the regulation of mRNA decapping in vivo. Although DCP2 still exhibits decapping activity in vitro, the decapping process of DCP2 is significantly inhibited in vivo. It should be noted that in vivo and in vitro experiments inherently measure different aspects of molecular function. The discrepancies between in vivo and in vitro experiments are of great significance because in vivo experiments more comprehensively reflect the regulatory role of DCP1. The in vitro environment lacks the complexity and numerous cofactors present in vivo, often failing to accurately replicate the interactions occurring within cells, highlighting the indispensable contribution of DCP1 in real mRNA decapping scenarios. Therefore, our study indicates that relying solely on in vitro experiments is insufficient to fully understand the function of DCP1; while important, in vitro experiments cannot completely capture the dynamics within cells. Together, these experiments deepen the understanding of DCP1's role in the decapping process in vivo. The lack of a significant difference in decapping activity observed in vitro does not negate DCP1's regulatory role in the cellular context; rather, it underscores our previous oversight of

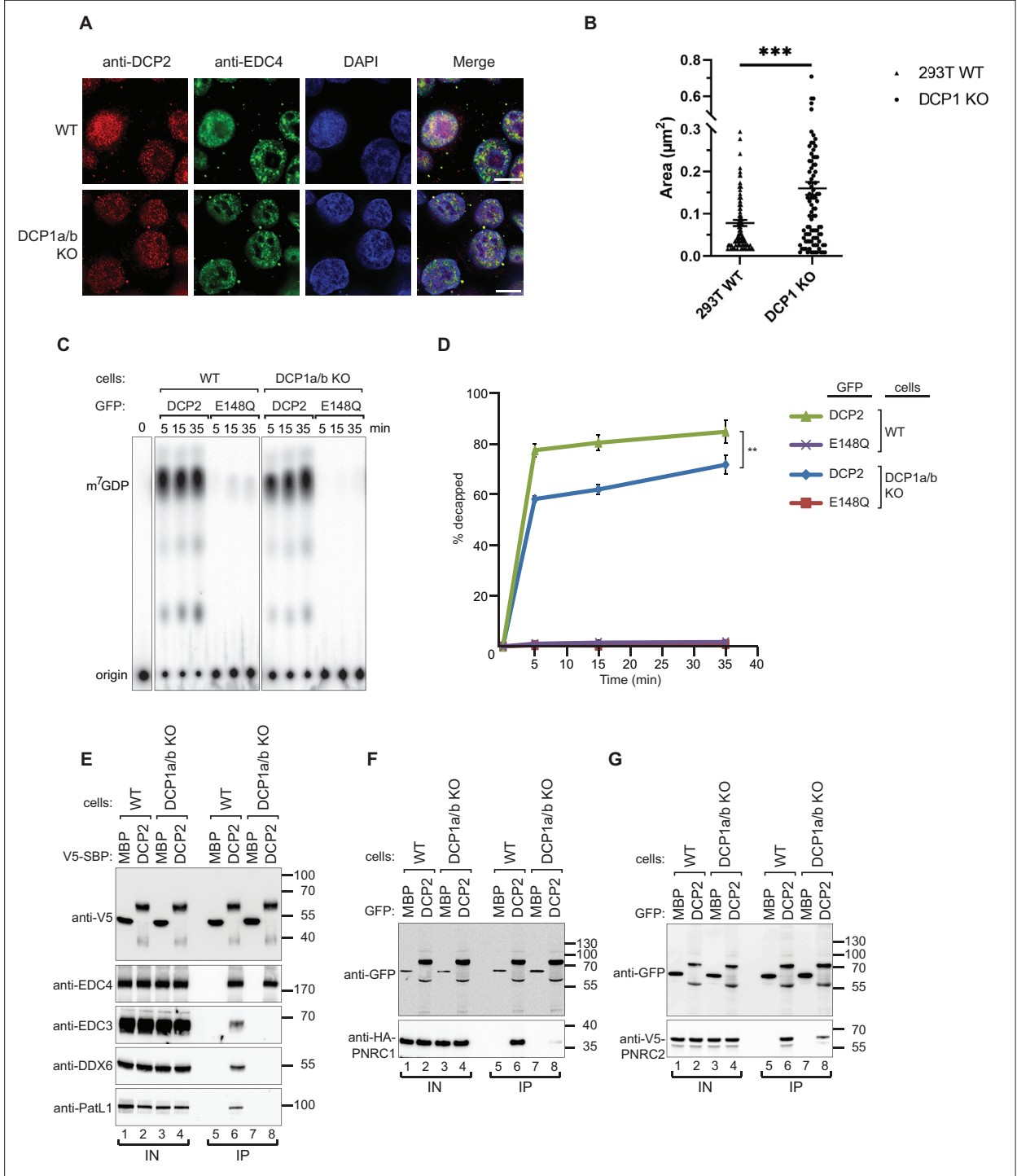

**Figure 2.** DCP1 serves as a bridging factor facilitating the interaction between multiple decapping factors and DCP2. (**A**) HEK-293T wild-type or DCP1a/b-null cells were stained with antibodies detecting DCP2 (red) and a P-body marker, EDC4 (green), and then counterstained with DAPI to visualize the nucleus (blue). In the merged image, colocalization of DCP2 and EDC4 appears yellow, indicating their interaction in P-bodies. This staining helps to visualize the cellular localization of DCP2 in the presence or absence of DCP1a/b. Scale bar = 10 μm. (**B**) Quantification of P-body size in wild-type and DCP1a/b-null HEK-293T cells. The average granule size was measured across at least three different fields of view. The middle line represents the mean of these measurements, and p-values were calculated using unpaired t-tests (ns: not significant; *p≤0.05; **p≤0.01; ***p≤0.001). (**C**) GFP-tagged DCP2 proteins were expressed in human HEK-293T wild-type or DCP1a/b-null cells. Following expression, the GFP-tagged DCP2 proteins were purified using GFP antibody and IgG beads. In vitro decapping activity was then tested, with the catalytically inactive DCP2 E148Q mutant serving as a negative control to demonstrate the specificity of the decapping activity. All experimental results were independently repeated at least three times.

*Figure 2 continued on next page*

*Figure 2 continued*

(**D**) Decapping assays in vitro were conducted to measure the fraction of decapped mRNA substrate by detecting the release of m$^7$GDP over time. The results are plotted as a function of time, with error bars representing the standard deviations (SD) from three independent experiments. An unpaired *t*-test was used to evaluate the statistical difference between samples (ns: not significant; *p≤0.05; **p≤0.01; ***p≤0.001). This panel demonstrates the decapping efficiency of DCP2 in the presence or absence of DCP1a/b. (**E**) V5-Streptavidin-Binding Peptide (SBP)-DCP2 proteins were expressed in human HEK-293T wild-type or DCP1a/b-null cells, followed by purification using Streptavidin beads. This experiment examines the interaction of V5-SBP-tagged DCP2 with endogenous decapping factors in wild-type or DCP1a/b knockout HEK-293T cells. The bound proteins were detected via western blot, with V5-SBP-MBP employed as a negative control. All experimental results were independently repeated at least three times. This panel highlights the role of DCP1a/b in facilitating or stabilizing interactions between DCP2 and other decapping factors. (**F, G**) The interaction of GFP-tagged DCP2 with HA-tagged PNRC1 (**F**) or V5-tagged PNRC2 (**G**) was assessed. The proteins were immunoprecipitated using anti-GFP antibodies and analyzed by western blotting with the indicated antibodies. All experimental results were independently repeated at least three times. These panels demonstrate the interaction between DCP2 and specific decapping co-factors, indicating how DCP1a/b may influence these interactions.

The online version of this article includes the following source data and figure supplement(s) for figure 2:

**Source data 1.** Original file for the decapping assays in *Figure 2C*.

**Source data 2.** *Figure 2C* and original scans of the relevant decapping assays with highlighted bands and sample labels.

**Source data 3.** Original file for the western blot in *Figure 2E*.

**Source data 4.** *Figure 2E* and original scans of the relevant western blot analysis with highlighted bands and sample labels.

**Source data 5.** Original file for the western blot in *Figure 2F*.

**Source data 6.** *Figure 2F* and original scans of the relevant western blot analysis with highlighted bands and sample labels.

**Source data 7.** Original file for the western blot in *Figure 2G*.

**Source data 8.** *Figure 2G* and original scans of the relevant western blot analysis with highlighted bands and sample labels.

**Figure supplement 1.** Colocalization of DCP2 with P-body markers in wild-type and DCP1a/b-null HEK-293T cells.

**Figure supplement 1—source data 1.** Original file for the western blot analysis in *Figure 2—figure supplement 1D*.

**Figure supplement 1—source data 2.** *Figure 2—figure supplement 1D* and original scans of the relevant western blot analysis with highlighted bands and sample labels.

DCP1's importance under in vitro conditions and the fact that DCP1's regulatory mechanisms in the decapping process remain to be elucidated.

## DCP1 is a central scaffold protein that recruits multiple decapping factors to DCP2

Since the discrepancies between in vivo and in vitro experiments reflect the regulatory role of DCP1 in decapping machinery. There are various factors and complex regulatory networks in the in vivo environment, which make it challenging to accurately simulate in vitro. To determine the underlying cause, we postulated that the complexity of the cellular setting, where multiple factors and cofactors are at play, contributes to these discrepancies. The observed cellular decapping defect in DCP1a/b knockout cells might be attributed to DCP1 functioning as a scaffold. It is conceivable that DCP1 facilitates the assembly of decapping modules, enabling their interaction with DCP2. Alternatively, this discrepancy could stem from the unique contributions of individual subunits, such as EDC4, EDC3, DDX6, and other decapping factors, to DCP2's activity. To explore this possibility, our investigation aimed to identify the specific decapping factors that rely on DCP1 for interaction with DCP2. We overexpressed V5-SBP- or GFP-tagged DCP2 in wild-type and DCP1a/b-knockout cells and then performed coimmunoprecipitation assays with RNase A treatment. In the absence of DCP1, interactions between DCP2 and the decapping factors EDC3, DDX6, and PatL1 (Edc3, Dhh1, and Pat1 in yeast, respectively) were almost completely suppressed, but not with EDC4 (also known as Ge-1 in *Drosophila*), a major component of the core decapping complex in metazoans (*Figure 2E*). Furthermore, the interaction between DCP2 and PNRC1, as well as PNRC2, which acts in synergy with DCP1 to promote decapping, was reduced in DCP1a/b-knockout cells (*Figure 2F and G*). These findings suggest that DCP1 acts as a binding platform for multiple decapping factors and is essential for the interaction of these cofactors with DCP2 in human cells.

## DCP2-mediated decapping is not merely due to the presence of decapping factors on the DCP1a scaffold

Based on our observation that DCP1a EVH1 domain is sufficient to rescue the decapping defect in DCP1a/b knockout cells, we hypothesized that the DCP1 EVH1 domain may play a key role in

regulating the decapping machinery. To evaluate the contribution of this domain to the formation of the decapping complex, we sought to identify which decapping factors interact with the DCP1 EVH1 domain. We overexpressed full-length or fragmented GFP-tagged DCP1a in DCP1a/b-knockout cells and subsequently immunoprecipitated the protein in the presence of RNase A. The EVH1 domain of DCP1a did not interact with any of the decapping factors except for DCP2 and PNRC1/2 (*Figure 3A–D*).

PNRC1 has been shown to interact with the NMD factor UPF1, indicating that it is involved in the NMD pathway (*Cho et al., 2009*). Furthermore, it has recently been shown that PNRC1 serves as a regulator of rRNA maturation within nucleoli, tightly controlling the process of ribosomal RNA maturation (*Gaviraghi et al., 2018*). Previous studies have also demonstrated the involvement of PNRC2 in the NMD pathway. Structural analysis of the PNRC2-DCP1a complex has shown that PNRC2 can directly activate the decapping process (*Cho et al., 2013*; *Cho et al., 2009*; *Lai et al., 2012*). Given this background, we investigated the functional role of PNRC1/2 in modulating the activity of DCP2 through DCP1 EVH1 domain.

To determine whether PNRC1 and PNRC2 are necessary for DCP2 activity in vivo, we used shRNA to knockdown the expression of PNRC1 and PNRC2 individually (*Figure 3—figure supplement 1*). Unlike the DCP1 knockout, the knockdown of PNRC1 or PNRC2 had only a minor effect on the results of the tethering assays, suggesting that PNRC1 and PNRC2 are not essential for DCP2 activity (*Figure 3E and F*). Therefore, we concluded the activation of the decapping complex by the DCP1a cannot be solely attributed to the binding of decapping factors on the DCP1a scaffold. This observation suggests that while DCP1's scaffolding function is crucial for recruiting cofactors, the decapping process likely involves additional layers of regulation that are not fully understood.

## DCP1 increases the mRNA-binding affinity of DCP2

Our findings suggest that the marked reduction in decapping activity observed in DCP1a/b-knockout cells is not exclusively essential for DCP1's function as a scaffold in the decapping process. This discrepancy prompts an investigation into the potential role of the DCP1 EVH1 domain, as previous studies have theorized its critical involvement in DCP2 mRNA binding (*Chang et al., 2014*; *Valkov et al., 2016*), although this aspect has not been explored to date. To evaluate this hypothesis, we investigated the molecular mechanisms of DCP1-mediated substrate binding by DCP2.

We evaluated the impact of DCP1 on interactions between DCP2 and mRNA in human cells using tethering assays with a catalytically inactive mutant DCP2 protein (DCP2 E148Q) introduced into DCP1a/b-knockout cells. This prevented the degradation of target mRNAs and maintained them in the decapping stage. Full-length DCP1a or the EVH1 domain alone were also introduced into the cells (*Figure 3G*). DCP2 samples were then immunoprecipitated and real-time PCR was used to measure the concentrations of reporter mRNAs that interacted with DCP2. Our results show that the mRNA binding affinity of DCP2 is weak in the absence of DCP1 but significantly increases in the presence of DCP1a full-length or the EVH1 domain (*Figure 3H and I*). While these changes may initially seem minor in vitro, their cumulative impact in the dynamic cellular environment could be substantial. Even minor perturbations in RNA binding affinity can trigger cascading effects, leading to significant changes in decapping activity and the accumulation of deadenylated intermediates upon DCP1 depletion. Cellular processes involve complex networks of interrelated events, and small molecular changes can result in amplified biological outcomes.

To further bolster the validity of these findings, we conducted RNA immunoprecipitation (RNA-IP) assays. Reporter RNA was isolated during the decapping process and consistent with the findings of the DCP2 immunoprecipitation assays, high levels of DCP2 were observed in the RNA immunoprecipitation assays when the DCP1a EVH1 domain was overexpressed (*Figure 3J*). These findings strongly underscore the significance of the EVH1 domain of DCP1a in enhancing interactions between DCP2 and mRNA, demonstrating how subtle molecular variations observed in vitro may translate into significant phenotypic outcomes within the complex cellular environment, emphasizing the critical regulatory role of DCP1a in the cellular decapping process. Taken together, we demonstrated that DCP1a can regulate DCP2's cellular decapping activity by enhancing DCP2's affinity to RNA, in addition to bridging the interactions of DCP2 with other decapping factors. This represents a pivotal molecular mechanism by which DCP1a exerts its regulatory control over the mRNA decapping process.

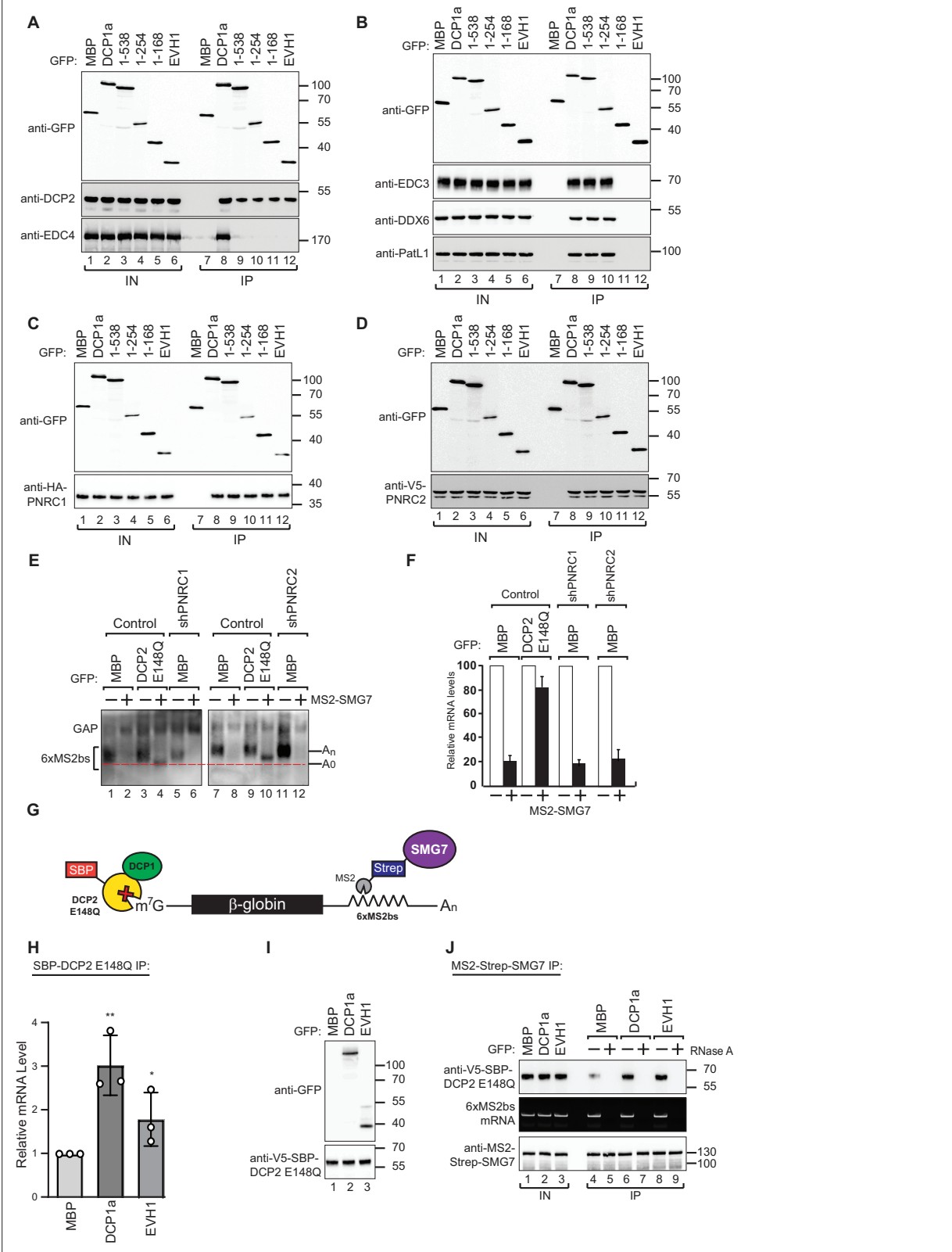

**Figure 3.** DCP1 facilitates DCP2 interactions with RNA molecules in human cells. (**A–D**) The interaction of GFP-tagged DCP2 with various decapping factors was examined. Proteins were immunoprecipitated using anti-GFP antibodies and analyzed by western blotting as described in **Figure 2D and E**. All experimental results were independently repeated at least three times. These panels demonstrate the binding affinity and specificity of DCP2 for different decapping co-factors, highlighting the importance of these interactions in the decapping process. (**E**) Tethering assays were performed in HEK-

*Figure 3 continued on next page*

*Figure 3 continued*

293T cells using control, shPNRC1, or shPNRC2 plasmids, following a procedure similar to that described in *Figure 1A*. All experimental results were independently repeated at least three times. This panel investigates the effect of knocking down PNRC1 or PNRC2 on the decapping activity, showing how these factors contribute to the stability and regulation of the β-globin-6xMS2bs mRNA. (**F**) The β-globin-6xMS2bs mRNA levels were normalized to those of the control mRNA. These normalized values were set to 100 in cells expressing MS2-HA (white bars). The mean values for relative mRNA levels in cells expressing MS2-SMG7 were estimated with standard deviations (SD) from three independent experiments (black bars). This quantification provides a comparative measure of the mRNA stability under different experimental conditions. (**G**) Schematic representation of the experimental procedure used in panels (**H**) and (**J**). This diagram outlines the steps and components involved in the transfection and subsequent analysis, providing a visual aid for understanding the experimental setup. (**H**) DCP1a/b-null HEK-293T cells were transfected with a mixture of plasmids, including β-globin-6xMS2bs, MS2-HA-Strep-SMG7, V5-SBP-DCP2 E148Q, and either full-length or the EVH1 domain of GFP-DCP1a. GFP-MBP was used as a control. The levels of β-globin-6xMS2bs mRNA bound to V5-SBP-DCP2 E148Q were immunoprecipitated using streptavidin beads and quantified by RT-PCR with GAPDH as a reference. The results represent three biological replicates' mean values ± SD. An unpaired *t*-test was used to evaluate the statistical difference between samples (ns: not significant; *p≤0.05; **p≤0.01; ***p≤0.001). This experiment assesses the role of different DCP1a constructs in the interaction and binding efficiency of DCP2 to the target mRNA. (**I**) The V5-SBP-DCP2 E148Q immunoprecipitated samples corresponding to panel (**H**) were analyzed by western blotting using the indicated antibodies. This panel confirms the tagged proteins' expression and proper immunoprecipitation, ensuring the RT-PCR results' validity in panel (**H**). (**J**) The plasmid mixture described in panel (**H**) was transfected into DCP1a/b-null HEK-293T cells. Subsequently, Strep-tag beads were used to immunoprecipitate the MS2-HA-Strep-SMG7 bound β-globin-6xMS2bs mRNA to study the in vivo interaction levels between RNA molecules and DCP2 E148Q in the presence of either full-length GFP-DCP1a or its EVH1 domain. GFP-MBP served as a control. All experimental results were independently repeated at least three times. This panel explores how different domains of DCP1a influence the interaction between DCP2 and its RNA targets, providing insights into the functional domains required for effective decapping.

The online version of this article includes the following source data and figure supplement(s) for figure 3:

**Source data 1.** Original file for the western blot in *Figure 3A*.

**Source data 2.** *Figure 3A* and original scans of the relevant western blot with highlighted bands and sample labels.

**Source data 3.** Original file for the western blot in *Figure 3B*.

**Source data 4.** *Figure 3B* and original scans of the relevant western blot analysis with highlighted bands and sample labels.

**Source data 5.** Original file for the western blot in *Figure 3C*.

**Source data 6.** *Figure 3C* and original scans of the relevant western blot analysis with highlighted bands and sample labels.

**Source data 7.** Original file for the western blot in *Figure 3D*.

**Source data 8.** *Figure 3D* and original scans of the relevant western blot analysis with highlighted bands and sample labels.

**Source data 9.** Original file for the northern blot analysis in *Figure 3E*.

**Source data 10.** *Figure 3E* and original scans of the relevant northern blot analysis with highlighted bands and sample labels.

**Source data 11.** Original file for the western blot in *Figure 3I*.

**Source data 12.** *Figure 3I* and original scans of the relevant western blot with highlighted bands and sample labels.

**Source data 13.** Original file for the western blot in *Figure 3J*.

**Source data 14.** *Figure 3J* and original scans of the relevant western blot with highlighted bands and sample labels.

**Figure supplement 1.** Validation of PNRC1 and PNRC2 knockdown in HEK-293T cells.

**Figure supplement 1—source data 1.** Original file for the western blot analysis in *Figure 3—figure supplement 1A*.

**Figure supplement 1—source data 2.** *Figure 3—figure supplement 1A* and original scans of the relevant western blot analysis with highlighted bands and sample labels.

**Figure supplement 1—source data 3.** Original file for the western blot analysis in *Figure 3—figure supplement 1B*.

**Figure supplement 1—source data 4.** *Figure 3—figure supplement 1B* and original scans of the relevant Western blot analysis with highlighted bands and sample labels.

## DCP1a and DCP1b regulate distinct endogenous downstream mRNA

The existence of two apparently redundant DCP1 paralogs, DCP1a and DCP1b, is intriguing, and raises the question whether they are truly redundant or in fact have distinct roles. Considering the low sequence similarity of DCP1a and DCP1b and since DCP1 was observed to enhance mRNA recognition by DCP2, we hypothesized that the DCP1 paralogs may contribute to the preferential regulation of specific mRNA targets. In other words, although DCP1a and DCP1b may be redundant in the general mRNA decapping process, they may differentially regulate specific mRNA targets.

We performed an RNA-seq analysis of HEK-293T cells lacking DCP1a, DCP1b, or both and compared the results with those from wild-type cells (*Figure 4A*, *Figure 4—figure supplement 1A–D*). There were significant differences in the transcript profiles of the knockout cells. In DCP1a-, DCP1b-, and

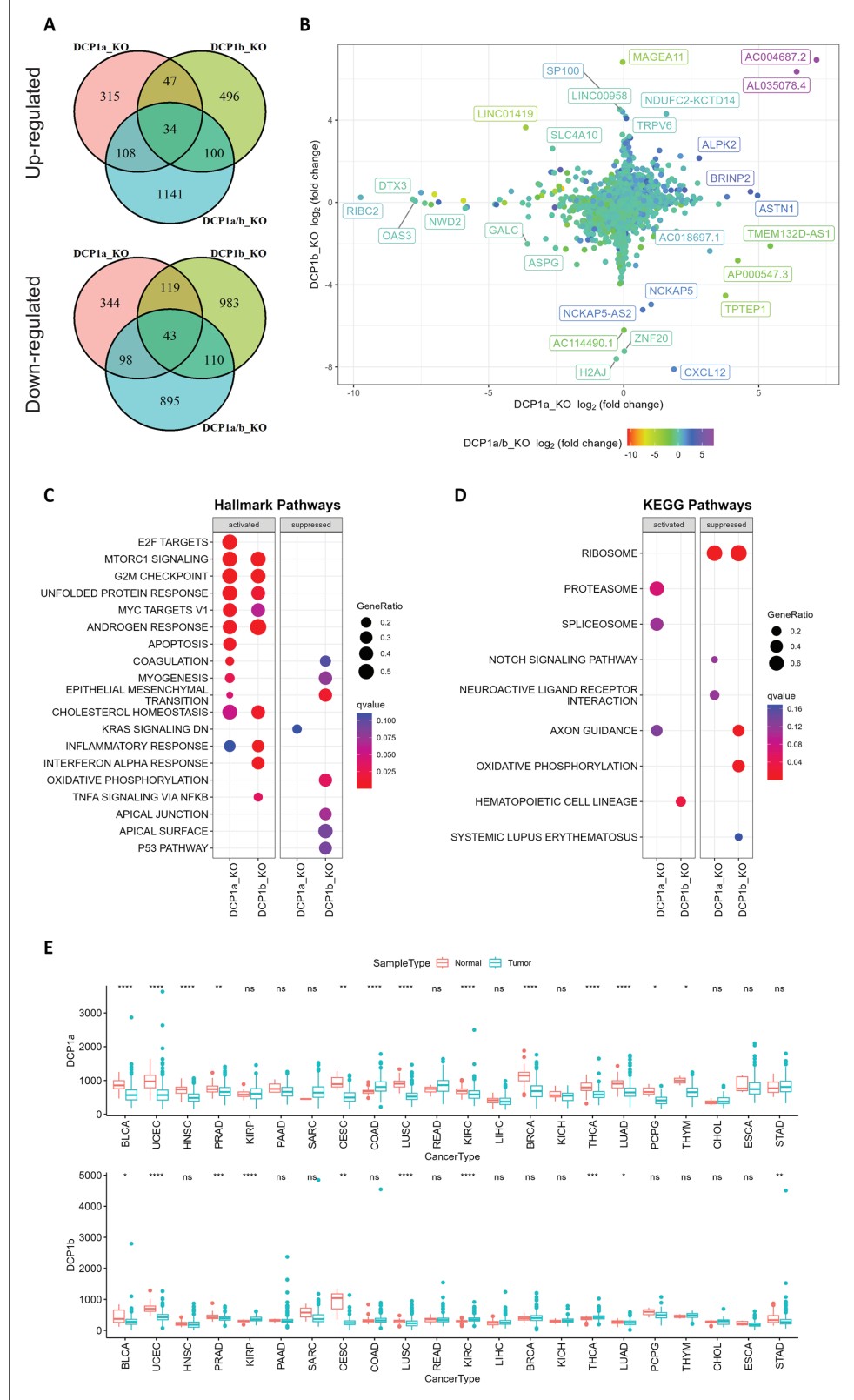

**Figure 4.** Gene expression analysis and pathway enrichment reveal distinct roles of DCP1a and DCP1b in human cells. (**A**) The upper and lower panels display Venn diagrams illustrating the number of genes that were significantly up- and downregulated in DCP1a, DCP1b, and DCP1a/b knockout cells (referred to as DCP1a_KO, DCP1b_KO, and DCP1a/b_KO), respectively. The overlapping regions indicate the number of genes significantly altered in

*Figure 4 continued on next page*

*Figure 4 continued*

multiple cell lines, highlighting shared and unique gene expression changes across different knockouts. (**B**) The plot shows the distribution of fold changes in gene expression when comparing DCP1a_KO or DCP1b_KO to wild-type (WT) cells. The points are colored according to the fold changes observed when comparing DCP1a/b_KO to WT, visually representing how the loss of DCP1a and DCP1b individually and combined affects gene expression. (**C**) Gene Set Enrichment Analysis (GSEA) results for hallmark pathways comparing DCP1a knockout and WT cells (left panel) and DCP1b knockout and WT cells (right panel). Activated and suppressed pathways are shown, with dots colored based on the q-value, indicating the statistical significance of the enrichment. (**D**) GSEA results for KEGG pathways comparing DCP1a knockout and WT cells (left panel) and DCP1b knockout and WT cells (right panel). Like panel (**C**), activated and suppressed pathways are shown, with dots colored according to the q-value. This analysis helps identify the biological pathways most affected by the loss of DCP1a or DCP1b. (**E**) Boxplots of the mRNA expression levels of DCP1a and DCP1b in various cancer types from The Cancer Genome Atlas (TCGA). Statistically significant differences in expression levels are labeled on top of each pair, with significance determined by the Wilcoxon test (ns: not significant; *p≤0.05; **p≤0.01; ***p≤0.001). This panel provides insights into the potential roles of DCP1a and DCP1b in different cancers, highlighting their differential expression patterns.

The online version of this article includes the following figure supplement(s) for figure 4:

**Figure supplement 1.** Transcriptome analysis and pathway enrichment in DCP1 knockout HEK-293T cells.

**Figure supplement 2.** Results of Cox regression analysis for survival analysis of genes DCP1a and DCP1b in progression-free interval (PFI) in cancers.

DCP1a/b-knockout cells, the expression of 604, 1255, and 1146 genes were significantly downregulated, while 504, 677, and 1383 genes were significantly upregulated (*Supplementary file 1*). Notably, only a subset of genes dysregulated in both DCP1a- and DCP1b-knockout cells showed alterations in DCP1a/b-knockout cells. This observation is further evidence of the crucial role of DCP1 in decapping (*Figure 4B*). It is worth mentioning that these significantly altered genes could be caused by both the direct effect of the accumulation of the target genes or the indirect effect of the accumulation of target genes. Furthermore, comparing the transcriptomes of the DCP1a- and DCP1b-knockout cells to the wild type results in many genes that are uniquely altered DCP1a-knockout or DCP1b knockout (*Figure 4A and B*), suggesting that these two proteins regulate distinct sets of genes, whether directly or indirectly. This observation implies that DCP1a and DCP1b are non-redundant in gene expression regulation.

## DCP1a and DCP1b play distinct roles in cancer and gene expression regulation

To identify the gene sets associated with DCP1a and DCP1b, we performed GSEA for Hallmark, Gene Ontology (GO), Kyoto Encyclopedia of Genes and Genomes (KEGG), and REACTOM pathways using the transcriptome data from DCP1a and DCP1b knockout experiments. Analysis of the Hallmark pathways revealed that multiple cancer-related pathways were activated in DCP1a-knockout and/or DCP1b-knockout cells, such as apoptosis, E2F targets, MYC targets, the mTOR signaling pathway, the G2/M checkpoint pathway, the KRAS signaling pathway, and the epithelial–mesenchymal transition pathway (*Figure 4C*). Results for the GO, KEGG, and REACTOM pathways likewise showed that DCP1a is involved in other cancer-related pathways, such as the Notch signaling pathway, embryonic skeletal system development, vasculature development, and the mitotic cell cycle pathway (*Figure 4D*, *Figure 4—figure supplement 1E–H*). These results are consistent with recent findings that DCP1a is strongly associated with embryonic growth and tumor development (*Ibayashi et al., 2021*; *Wu et al., 2018*; *Wu et al., 2021b*), and suggest that DCP1a is more strongly associated with cancer than is DCP1b. Furthermore, we noted that the activation of capped intron-containing pre-mRNA processing was evident in both DCP1a- and DCP1b-deficient cells (*Figure 4—figure supplement 1H*). This observation aligns with the understanding that the depletion of DCP1a not only diminishes mRNA degradation and transcription but also underscores the pivotal role of DCP1 in the orchestration of gene expression. To further assess the clinical relevance of our findings, we evaluated the expression levels of DCP1a and DCP1b using data from The Cancer Genome Atlas (TCGA) database (*Figure 4E*). Consistent with the above results, we found that expression levels of DCP1a were reduced in various cancers: bladder urothelial carcinoma, uterine corpus endometrial carcinoma, head and neck squamous cell carcinoma, prostate adenocarcinoma, cervical squamous cell carcinoma

and endocervical adenocarcinoma, lung squamous cell carcinoma, kidney renal clear cell carcinoma, breast invasive carcinoma, thyroid carcinoma, lung adenocarcinoma, pheochromocytoma and paraganglioma, and thymoma. In contrast, DCP1b exhibited significant upregulation in certain cancer types while showing downregulation in others. Additionally, higher expression levels of DCP1a are found to be associated with a longer progression-free interval (PFI) in uterine carcinosarcoma carcinoma (*Figure 4—figure supplement 2A*). Notably, higher expression of DCP1b is associated with a shorter PFI in brain lower grade glioma (*Figure 4—figure supplement 2B*) and associated with a longer PFI in many other cancer types, including adrenocortical carcinoma, prostate adenocarcinoma, uveal melanoma, pancreatic adenocarcinoma, and uterine corpus endometrial carcinoma (*Figure 4—figure supplement 2C*). This suggests that the role of DCP1b in tumorigenesis is multifaceted and context-dependent. In summary, these findings indicate that DCP1a and DCP1b likely have distinct targets and play non-redundant roles. The nuanced functions of DCP1a and DCP1b warrant further investigation to fully elucidate their specific contributions.

## DCP1a and DCP1b play different roles in cellular metabolism

To decipher the distinct biological functions of DCP1a and DCP1b, we employ metabolic profiling to investigate the complex metabolic alterations resulting from their absence, thus deepening our grasp of their cellular roles. Through untargeted metabolomic profiling of DCP1a-, DCP1b-, and

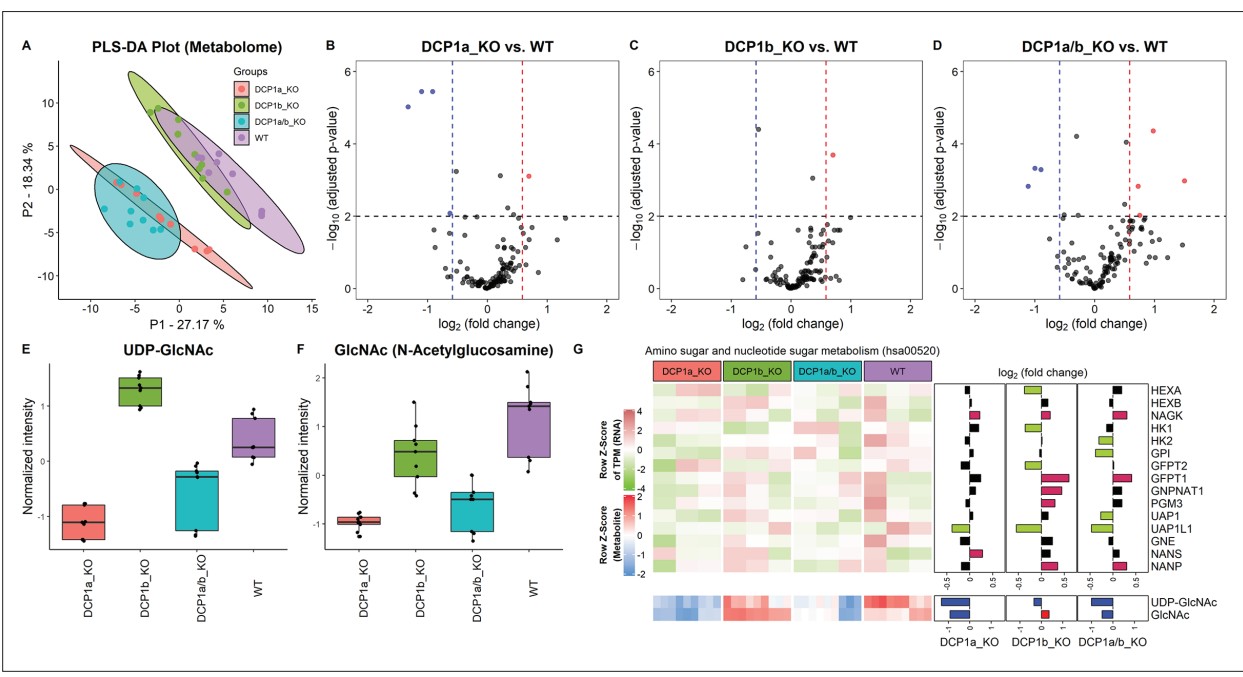

**Figure 5.** Metabolome profiling unveils unique functions of DCP1 paralogs in human cells. (**A**) Partial least squares discriminant analysis (PLS-DA) plot illustrating the metabolome profiles of 12 samples, with three technical replicates for each knockout cell line. This plot shows how the metabolomic profiles of DCP1a_KO, DCP1b_KO, and DCP1a/b_KO cells differ from each other and wild-type (WT) cells, indicating distinct metabolic alterations resulting from the knockouts. (**B–D**) Volcano plots comparing metabolite levels between (**B**) DCP1a_KO and WT, (**C**) DCP1b_KO and WT, and (**D**) DCP1a/b_KO and WT. Metabolites with an absolute fold change greater than 1.5 and a false discovery rate (FDR)-adjusted p-value less than 0.01 were considered significant. These plots highlight the metabolites significantly upregulated or downregulated in each knockout compared to WT, providing insight into the metabolic impact of DCP1a and DCP1b loss. (**E, F**) Box plots of (**E**) UDP-GlcNAc and (**F**) GlcNAc levels between different cell lines. These plots show the relative abundance of specific metabolites in the different knockout and WT cells, indicating how the disruption of DCP1a and DCP1b affects specific metabolic pathways. (**G**) Normalized expression levels, abundance of transcripts and metabolites, and fold changes in the amino sugar and nucleotide sugar metabolism pathway. Significantly upregulated and downregulated genes are colored in pink and blue, respectively. Upregulated and downregulated metabolites are colored in red and blue, respectively. This panel integrates transcriptomic and metabolomic data to provide a comprehensive view of how the amino sugar and nucleotide sugar metabolism pathway is altered in DCP1 knockout cells.

The online version of this article includes the following figure supplement(s) for figure 5:

**Figure supplement 1.** Metabolomic analysis of DCP1 knockout and wild-type HEK-293T cells.

**Figure supplement 2.** Amino sugar and nucleotide sugar metabolism pathways in DCP1 knockout HEK-293T cells.

DCP1a/b-knockout cells, we aim to thoroughly assess the impact of these knockouts, uncovering the unique roles of these proteins. After removing features that appeared in control samples and manually removing features with a poor peak shape and extremely low abundance, we identified approximately 123 metabolites in the experimental samples. We successfully differentiated DCP1a- and DCP1b-knockout cells using PLS-DA score plots, loading plots, and heatmaps of the metabolomic data and the differences in the metabolomic profiles revealed the distinct impacts of DCP1a and DCP1b on cellular metabolism (*Figure 5A*, *Figure 5—figure supplement 1A and B*). Additionally, the volcano plots illustrated the significant metabolic disparities among DCP1a-, DCP1b-, and DCP1a/b-knockout cells (*Figure 5B–D*), emphasizing the presence of noticeable differences. Remarkably, uridine diphosphate (UDP)-N-acetylglucosamine (GlcNAc), GlcNAc, glycerophosphoserine, and glycerophosphocholine levels were significantly elevated in DCP1a- and DCP1a/b-knockout cells, whereas those of nicotinamide adenine dinucleotide (NAD) were reduced. By contrast, DCP1b-knockout cells had elevated levels of lysophosphatidylethanolamine (P-18:0) [LPE (P-18:0)] (*Figure 5E and F*, *Figure 5—figure supplement 1C*). These results suggest that DCP1a and DCP1b have different effects on these metabolic pathways.

The significantly altered metabolites in the DCP1-knockout cells included UDP–GlcNAc and GlcNAc, which are associated with the amino sugar and nucleotide sugar metabolism pathway (hsa00520) in KEGG (*Figure 5—figure supplement 2*). Further investigation revealed that the levels of these metabolites are correlated with the transcript levels associated with the amino sugar and nucleotide sugar metabolism pathway (*Figure 5G*). Together, our data strongly imply that DCP1a and DCP1b may exert regulatory influence over UDP-GlcNAc and GlcNAc through pathways that extend beyond the scope of amino sugar and nucleotide sugar metabolism. This indicates that these proteins play intricate and far-reaching roles in modulating cellular metabolism. Further studies are necessary to unravel the specific mechanisms and pathways by which DCP1a and DCP1b control these metabolites and to fully elucidate their roles in cellular metabolism.

## Discussion

DCP1 is a key component in the mRNA decay pathway, crucial for the regulation of gene expression. Research across various species, from yeast to humans, has provided insights into the conservation and evolution of mRNA decapping mechanisms. Notably, DCP1 interacts with other decapping proteins such as DCP2, forming a decapping complex for mRNA degradation. Additionally, the localization and functional analysis of DCP1 in different model organisms, as well as its importance in plant species, highlight its role in post-transcriptional gene expression regulation across species (*Lall et al., 2005*; *Sakuno et al., 2004*; *Xu et al., 2006*). However, in human cells, the specific biological functions of DCP1, especially the distinctions between its two paralogs, DCP1a and DCP1b, have not been fully elucidated. In this study, we found that DCP1 plays a key role in promoting mRNA decapping in human cells. By generating cell lines deficient in DCP1a, DCP1b, or both, our research emphasizes the importance of human DCP1 in the decapping process. We also verified that DCP1 is not only a key platform for assembling various mRNA decapping cofactors, but also established that the EVH1 domain of DCP1 is critical for the mRNA decapping process. Through RNA immunoprecipitation assays, we discovered that the interactions between DCP2 and mRNA heavily rely on the presence of DCP1's EVH1 domain. These findings imply that DCP1 facilitates the binding of DCP2 to target mRNAs through multiple mechanisms, thereby enhancing the efficiency of mRNA decapping. Finally, our integrated transcriptome and metabolome analyses comparing the specific functions of DCP1a and DCP1b in human cells revealed that they can regulate distinct endogenous mRNA targets and biological processes.

In a prior investigation, we made an intriguing discovery. In human cells, substituting the conserved NR-loop with a flexible GSSG loop in the EVH1 domain of DCP1 had no adverse effect on the interaction between DCP1 and DCP2, nor did it impede their integration into decapping factors. However, this seemingly subtle alteration had a significant consequence – it disrupted the activation of DCP2 (*Chang et al., 2014*). This points to the pivotal role played by the EVH1 domain of DCP1 in the activation of DCP2, which, in turn, is a critical step in the mRNA decapping process. These findings, combined with our findings in the present study, provide compelling evidence that the DCP1 EVH1 domain is not just a passive component but a crucial player in coordinating interactions and facilitating conformational changes necessary for efficient mRNA decapping. A comprehensive functional

characterization of these mechanisms is imperative to fully elucidate the mechanisms underlying DCP1-mediated activation of decapping.

While the protein components of the mRNA decapping machinery have been identified, our current understanding of the distinct roles played by individual decapping factors throughout the process remains limited. Notably, whether the different decapping factors confer target specificity has yet to be clarified. Although previous studies have suggested that DCP1a and DCP1b, the two paralogs of DCP1, play redundant roles in general mRNA decay, this notion is mainly based on reporter assays and single-gene approaches. The specific regulation of endogenous mRNA targets and biological processes by DCP1a and DCP1b is yet to be elucidated.

In this study, we investigated potential functional differences between DCP1a and DCP1b through a combined transcriptome and metabolome analysis. While DCP1a and DCP1b have both been implicated in cancer, our observation of significant alterations in multiple cancer-related pathways in DCP1a-knockout cells suggests that DCP1a might be more prominently associated with cancer development. Furthermore, the downregulation of DCP1a in multiple cancer types further supports the concept that DCP1a is a cancer-associated gene and suggests that its dysregulation may play a contributory role in the onset and progression of diverse cancers. (*Ibayashi et al., 2021*; *Wu et al., 2018*; *Wu et al., 2021b*). Survival analysis has indicated that elevated DCP1a expression in uterine carcinosarcoma carcinoma correlates with improved PFI. Conversely, higher levels of DCP1b expression are associated with both favorable and unfavorable PFIs. Together with the observation of dysregulation of DCP1b in numerous cancers, we speculate that DCP1b is also implicated in cancer pathogenesis, and further studies are required to fully understand the roles of DCP1a and DCP1b in cancer pathogenesis.

With the transcriptome profiles of DCP1a and DCP1b knockout cells, we anticipate significant upregulation among potential targets. However, to discern direct targets from genes upregulated due to indirect effects, it is imperative to delve deeper into mRNA decay rates. Various tools and methods, such as time-series RNA-seq (*Viegas et al., 2023*) or specialized sequencing techniques like Precision Run-On sequencing (PRO-seq) (*Blumberg et al., 2021*) or RNA Approach to Equilibrium Sequencing (RATE-Seq) (*Abdul-Rahman and Gresham, 2018*), have been proposed to estimate mRNA decay rates. Luo et al. utilized TimeLapse-seq to identify potential targets for DCP2 (*Luo et al., 2021*; *Schofield et al., 2018*). Incorporating these approaches could provide a more comprehensive understanding in this context.

Notably, our candidate target genes exhibited minimal overlap with the 91 targets of DCP2 (*Luo et al., 2021*); only EPPK1 overlapped among potential targets of DCP1b_KO, while TES, PAX6, and C18orf21 were the only overlapping genes in the DCP1a/b double knockout.

Our scrutiny of the transcript profiles and metabolite levels within DCP1a- and DCP1b-knockout cells has unveiled their unique contributions to cellular processes. While DCP1a and DCP1b might exhibit certain common gene targets, the distinct transcriptome profiles suggest their functions are not entirely interchangeable. Pathway analysis indicated that DCP1a and DCP1b may have partially complementary roles in regulating cellular pathways. Additionally, several metabolites showed notable alterations in knockout cells, potentially opening a new avenue for investigating the metabolic consequences of DCP1 knockout. Our findings yield valuable understanding of the distinct functions of DCP1a and DCP1b in human cells and the complex regulatory mechanisms underlying cellular metabolism. The potential targets we identified could be investigated further by quantifying specific transcripts and metabolites, and a larger sample size would facilitate the identification of correlations between differentially expressed genes and differentially abundant metabolites.

Overall, our study provides valuable insights into the molecular mechanisms governing DCP1-mediated decapping in human cells and the coordinated regulation of DCP1a and DCP1b, which has important implications for gene expression and cellular function. Future biochemical characterization of the specific mRNA targets and biological pathways regulated by DCP1a and DCP1b is necessary to fully elucidate their roles. This information may help uncover the precise mechanisms underlying mRNA decapping and gene regulation. Furthermore, studying the dysregulation of DCP1 paralogs, especially in cancer, may reveal potential therapeutic targets and improve treatment strategies.

## Materials and methods

### DNA constructs

The DNA constructs used in this study are listed in Appendix 1. The plasmids for expression of the β-globin-6xMS2bs and the control β-globin-GAPDH (control) mRNAs were kindly provided by Dr. J. Lykke-Andersen and have been described previously (*Lykke-Andersen et al., 2000*).

### Cell culture and transfections

Human HEK-293T cells were purchased from ATCC (CRL-3216) and were cultured in DMEM (Gibco 11995) supplemented with 10% (v/v) fetal bovine serum (Gibco) and grown at 37°C with 5% $CO_2$. The identity of these cells was authenticated through SRT profiling, and they were confirmed to be negative for mycoplasma.

### CRISPR/Cas9-mediated gene editing

The HEK-293T DCP1a-, DCP1b-, and DCP1a/b-null cell lines were generated by CRISPR/Cas9-mediated gene editing as described previously (*Sgromo et al., 2018*). The guide RNA targeting human DCP1 paralog genes (DCP1a: GCTGTTTCGAGGTAAGCTGG; DCP1b: TCGGCCATCGGG CCAACGAG) was designed using the ATUM CRISPR gRNA Design tool. Genomic DNA was isolated from single clones using the Wizard SV Genomic DNA Purification System (Promega). The DCP1a/DCP1b locus was PCR amplified, and Sanger sequencing of the targeted genomic regions confirmed the presence of a deletion within the targeted exon, resulting in a frameshift. The absence of detectable levels of DCP1a or DCP1b protein was confirmed by western blotting.

### Tethering assays

Tethering assays using the MS2 reporter system were performed as described previously (*Chang et al., 2014*). Briefly, HEK-293T cells were cultured in six-well plates and transiently transfected with a mixture of four plasmids: 0.5 µg control plasmid (β-globin-GAPDH), 0.5 µg plasmid encoding the β-globin-6xMS2bs, 0.5 µg plasmids encoding the MS2-tagged fusion protein (SMG7, TNRC6A-SD [silencing domain], or Nanos1) and plasmids encoding GFP-MBP, GFP-DCP2 E148Q, or GFP-DCP1 (wild-type or fragments) as indicated amounts. The cells were harvested 2 days after transfection. Total RNA was isolated using the TRIzol method (Thermo), separated on a denaturing agarose gel, transferred onto a nylon membrane (Amersham Hybond-N⁺), and hybridized with a probe synthesized in vitro using the full-length β-globin DNA sequence labeled with $^{32}$P. The hybridized membrane was then analyzed by northern blot.

### 5'-Phosphate-dependent exonuclease assay

The integrity of the 7-methylguanosine cap structure at the 5'-ends of the transcripts was investigated in a 5'-phosphate-dependent exonuclease assay. 10 µg RNA extracted from the specific tethering assays was incubated in a 20 µl reaction volume with 1 unit of Terminator 5'-phosphate-dependent exonuclease (Epicentre) for 60 min at 30°C. The Terminator was omitted in the control. The reaction was then stopped by adding phenol. After standard extraction and ethanol precipitation, RNA levels were visualized by northern blotting.

### Immunofluorescence

Cells were fixed and permeabilized as described previously (*Jakymiw et al., 2005*). The antibodies used in this study are listed in Appendix 1.

### Decapping assays

Decapping assays were performed with immunoprecipitated wildtype GFP-DCP2 or a catalytic mutant (E148Q) from either wild-type DCP1a/b-null HEK-293T cells, along with in vitro-synthesized RNA (127 nucleotides) with a $^{32}$P-labeled cap structure (*Chang et al., 2014*). To generate the $^{32}$P-labeled cap RNA, we utilized the ScriptCap m$^7$G Capping System and ScriptCap 2'-O-methyltransferase kit (Epicentre Biotechnologies) with [α-$^{32}$P] GTP for our experiments. The concentration of RNA used was 0.5 µM. The decapping reactions were performed at 30°C for the indicated times in a total volume of 10 µl of decapping buffer (50 mM Tris-HCl [pH 7.5], 50 mM ammonium sulfate, 0.1% [w/v] BSA, and

5 mM MgCl$_2$). The proteins were diluted to their working concentrations with the decapping buffer. The reactions were stopped by adding up to 50 mM EDTA (final concentration), and 1 µl of each sample was spotted on polyethylenimine (PEI) cellulose thin-layer chromatography plates (Merck) and developed in 0.75 m LiCl.

## Co-immunoprecipitation assays and western blot analysis

For immunoprecipitation assays, HEK-293T cells were seeded in 10 cm dishes and transfected with 20 µg total plasmid DNA using Lipofectamine 2000 (Thermo). The cells were washed 48 hr after transfection with phosphate-buffered saline and lysed in 1 ml NET buffer (50 mM Tris-HCl [pH 7.5], 150 mM NaCl, 1 mM EDTA, 0.1% [v/v] Triton-X-100, 10% [v/v] glycerol and supplemented with complete protease inhibitor cocktail [Sigma]). Immunoprecipitations were performed as described previously (*Braun et al., 2012*). RNase A (1 µl, 10 mg/ml) was added to the cell extract in all experiments. The antibodies used in this study are listed in Appendix 1. All western blots were developed with the ECL Western blotting analysis system (GE Healthcare) as recommended by the manufacturer.

## PNRC1 and PNRC2 knockdown

Plasmids expressing short-hairpin RNAs (shRNAs) for knockdowns were derived from the pSUPER plasmid containing the puromycin-resistance gene for selection. The vector backbone was a kind gift from O. Mühlemann (University of Bern). The 19 nt target sequences were as follows: control: ATTCTCCGAACGTGTCACG, PNRC1: CAAAGTTTAGTGATCCACCTTT, and PNRC2: AGTTGGAA TTCTAGCTTAT. HEK-293T cells were grown in DMEM supplemented with 10% heat-inactivated fetal bovine serum and 2 mM L-glutamine. The cells were transfected in six-well plates using Lipofectamine 2000 (Thermo) according to the manufacturer's protocol. Transfection mixtures contained 0.5 µg control plasmid (β-globin-GAPDH), 0.5 µg plasmid encoding the β-globin-6xMS2bs, 0.5 µg plasmids encoding the MS2-tagged SMG7, and 2.5 µg of plasmids expressing the relevant shRNA. Then, 24 hr after transfection, cells were selected in medium supplemented with 1.5 µg/ml puromycin. After 24 hr of selection, cells were counted and reseeded in new six-well plates in medium without puromycin for recovery. After 24 hr of reseeding, the cells were prepared for re-transfection to conduct tethering assays.

## RNA granule quantification

Quantification of P-bodies was performed using Fiji/ImageJ software. Nuclei were counted in the DAPI channel utilizing the cell counter plugin. P-bodies in the 488 channel (stained with EDC4 for P-bodies) were detected using a binary threshold and primary object detection for objects between 40 and 255 pixels. P-body sizes were quantified using the 'Analyze Particles' function, with detection parameters set to sizes between 0.00 and infinity (inch²) and circularity between 0.00 and 1.00. Size and intensity thresholds were maintained consistently across experiments with identical staining parameters. Excluding P-bodies on image edges, all results are presented as means ± standard errors of the means (SEMs) from experiments repeated independently at least three times. Unpaired *t*-tests were used to evaluate statistical differences between samples. Quantification of P-body size in wild-type and DCP1a/b-null HEK-293T cells was conducted, measuring average granule size across at least three fields of view. The middle line represents the mean of these measurements, with p-values calculated using unpaired *t*-tests (*p≤0.05; **p≤0.01; ***p≤0.001).

## RNA immunoprecipitation assays

To perform RNA immunoprecipitation assays, HEK-293T DCP1a/b-null cells were cultured in 10 cm dishes and transiently transfected with a mixture of four plasmids: 5 µg of β-globin-6xMS2bs, 5 µg of plasmids encoding the MS2-HA-Strep-tagged SMG7, 5 µg of plasmids encoding V5-Streptavidin-Binding Peptide (SBP)-tagged DCP2 E148Q, and 5 µg of plasmids encoding GFP-MBP. Then, 48 hr after transfection, V5-SBP-DCP2 E148Q or MS2-HA-Strep-SMG7 was immunoprecipitated by Strepta-vidin (Thermo) or Strep-Tactin (IBA) beads, respectively, as described previously (*Braun et al., 2012*).

The level of DCP2-bound reporter mRNA was extracted and reverse transcribed using b-globin reverse primer (TTAGTGATACTTGTGGGCCAGGGC). The amount of target mRNA was subsequently determined using quantitative PCR (qPCR) using the respective primer pair (Fw: ATGGTGCACCTG ACTCCTGAG; Rev: TTAGTGATACTTGTGGGCCAGGGC). To use the Livak method (ΔΔCT method)

for relative quantification, each primer pair was tested if the amplification rate of the specific PCR product was 2 ± 5%. To determine the level of DCP2 bound to reporter mRNA, Strep-tagged SMG7 was precipitated using Strep-Tactin beads, and the DCP2 level was detected by western blot.

## Transcriptome sequencing (RNA-seq)

HEK-293T wild-type, DCP1a-, DCP1b-, or DCP1a/b-null cells were plated on 15 cm dishes 24 hr before harvesting as described previously (*Calviello et al., 2016*). Total RNA was extracted using the RNeasy Mini Kit (QIAGEN) and a library prepared using the TruSeq RNA Sample Prep Kit (Illumina). Three biological replicates were analyzed. RNA-seq libraries were sequenced using the Illumina NovaSeq 6000 sequencing system.

## Metabolomics analysis (HILIC, LC–QToF)

LC-MS grade methanol and water were procured from Scharlau Chemie (Sentmenat, Barcelona, Spain) and MS-grade acetonitrile was obtained from J.T. Baker (Phillipsburg, NJ). The remaining chemicals were purchased from MilliporeSigma (St. Louis, MO), unless specified otherwise.

The number of cultured cells in each tube was adjusted to $10^6$ in 100 µl PBS and quenched by using 400 µl of cold methanol. After centrifuging at 15,000 × *g* for 5 min, the supernatant was collected and stored at –20°C until use. Metabolomics data were acquired using an Agilent 1290 ultra-high-performance liquid chromatography system (Agilent Technologies, Waldbronn, Germany) connected to a Bruker maXis ultra-high-resolution (UHR)-time-of-flight (TOF) mass spectrometer (Bruker Daltonics, Bremen, Germany). Metabolites were separated using a BEH Amide column (2.1 mm × 100 mm, 1.7 µm), with an injection volume of 10 µl. The autosampler and column temperatures were maintained at 4 and 40°C, respectively. The mobile phase A consisted of 10 mM ammonium acetate in deionized water with 0.1% formic acid, and the mobile phase B consisted of 10 mM ammonium acetate in a 5:95 (v/v) water:acetonitrile mixture with 0.1% formic acid. The flow rate was 0.4 ml min$^{-1}$. The elution gradient was as follows: 0–0.5 min, 99% mobile phase B; 0.5–7 min, 99%–50% mobile phase B; 7–10 min, 50% mobile phase B; and column re-equilibration with 99% mobile phase B for 2 min. The electrospray settings were: dry gas temperature, 200°C; dry gas flow rate, 8 l min$^{-1}$; nebulizer gas pressure, 2 bar; capillary voltage, 4500 V; endplate offset potential, 500 V. Mass spectra were recorded over the range 50–1500 *m/z* in the positive mode. The TOF mass analyzer was calibrated using sodium formate with a mass range of 50–1500 *m/z* before use.

The maXis UHR-TOF data were processed using MS-DIAL version 4.60 (accessible at http://prime.psc.riken.jp/) and Bruker Compass DataAnalysis version 4.1. For peak detection and alignment, the minimum peak height was set at an amplitude of 3000, the retention time tolerance was set at 0.1 min, and other parameters were set at their default values using the web-based metabolomics data processing tool MetaboAnalyst 5.0 (accessible at https://www.metaboanalyst.ca/). For statistical analysis, the data were normalized (mean-centered and divided by the standard deviation of each variable) and heat maps and volcano plots were constructed. Partial least squares discriminant analysis (PLS-DA) was performed using the R package, mixOmics (*Rohart et al., 2017*). Known features were first confirmed using a home-built compound library generated using commercial reference standards based on retention time and isotopic mass information. Unknown features of interest were further identified using the online databases METLIN (*Guijas et al., 2018*) and the Human Metabolome Database (*Wishart et al., 2018*).

## Transcriptomics analysis

Transcriptomes were sequenced using high-throughput technology. Low-quality bases and adapters were removed with Trimmomatic (*Bolger et al., 2014*). The filtered, high-quality reads were mapped to the human genome (GENCODE Human Release 33) and quantified with RSEM (*Frankish et al., 2019*; *Li and Dewey, 2011*). The TPM expression profiles from each sample were then used to generate PCA plots with R. The fold changes between different groups for each gene were estimated with DESeq2 (*Love et al., 2014*). Genes were defined as differentially expressed if the absolute value of the corresponding fold change was larger than 1.5 and the adjusted p-value was less than 0.05. Pathway analysis was performed using the clusterProfiler R package (*Wu et al., 2021a*); specifically, GSEA (*Subramanian et al., 2005*) was used to identify activated or suppressed GO terms (*The Gene Ontology, 2019*; *The Gene Ontology, 2019*), REACTOME pathways (*Croft et al., 2011*), and KEGG

pathways (*Kanehisa and Goto, 2000*). All the plots were generated with in-house R scripts using the packages ggplot2 and ComplexHeatmap (*Gu et al., 2016*; *Wickham, 2016*). The expression levels of DCP1a and DCP1b for different cancer types in TCGA were download from PanCanAtlas (*Weinstein et al., 2013*). Survival analysis were carried out with webserver DoSurvive (*Wu et al., 2023*).

## Acknowledgements

We dedicate this work to Elisa Izaurralde, who passed away during the study period. We thank Sigrun Helms and Heike Budde for their technical assistance. We thank the Metabolomics Core Laboratory, Centers of Genomic and Precision Medicine, National Taiwan University for the instrumental support.

## Additional information

### Funding

| Funder | Grant reference number | Author |
| --- | --- | --- |
| National Science and Technology Council | MOST109-2311-B-010-001-MY2 | Chung-Te Chang |
| Yen Tjing Ling Medical Foundation | CI-110-17 | Chung-Te Chang |

The funders had no role in study design, data collection and interpretation, or the decision to submit the work for publication. Open access funding provided by Max Planck Society.

### Author contributions

Ting-Wen Chen, Data curation, Software, Formal analysis, Visualization, Methodology, Writing - original draft, Writing - review and editing; Hsiao-Wei Liao, Resources, Data curation, Software, Formal analysis, Investigation, Visualization, Methodology, Writing - original draft, Writing - review and editing; Michelle Noble, Resources, Data curation, Formal analysis, Validation, Investigation, Visualization, Writing - review and editing; Jing-Yi Siao, Data curation, Formal analysis, Validation, Visualization, Writing - original draft; Yu-Hsuan Cheng, Formal analysis, Validation, Visualization, Writing - review and editing; Wei-Chung Chiang, Resources, Data curation, Software, Methodology; Yi-Tzu Lo, Formal analysis, Visualization; Chung-Te Chang, Conceptualization, Resources, Supervision, Funding acquisition, Investigation, Methodology, Writing - original draft, Project administration, Writing - review and editing

### Author ORCIDs

Chung-Te Chang ⬥ https://orcid.org/0000-0002-4792-1646

Reviewer #1 (Public review): https://doi.org/10.7554/eLife.94811.3.sa1
Author response https://doi.org/10.7554/eLife.94811.3.sa2

## Additional files

### Supplementary files

• Supplementary file 1. Differentially expressed genes identified by comparing transcriptomes between DCP1 knockout and wild-type samples. (**a**) Differentially expressed genes identified by comparing transcriptomes between DCP1a knockout and wild-type samples. (**b**) Differentially expressed genes identified by comparing transcriptomes between DCP1b knockout and wild-type samples. (**c**) Differentially expressed genes identified by comparing transcriptomes between DCP1a/b knockout and wild-type samples.

• MDAR checklist

### Data availability

Sequencing data have been deposited in GEO under accession codes GSE230847.

The following dataset was generated:

| Author(s) | Year | Dataset title | Dataset URL | Database and Identifier |
|---|---|---|---|---|
| Chen TW, Chang CT | 2023 | Uncovering human DCP1 is essential for mRNA decapping process and paralog-specific functions in gene regulation | https://www.ncbi.nlm.nih.gov/geo/query/acc.cgi?acc=GSE230847 | NCBI Gene Expression Omnibus, GSE230847 |

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

# Appendix 1

**Appendix 1—key resources table**

| Reagent type (species) or resource | Designation | Source or reference | Identifiers | Additional information |
|---|---|---|---|---|
| Gene (*Homo sapiens*) | DCP1a | GenBank | HGNC:18714 | |
| Gene (*H. sapiens*) | DCP1b | GenBank | HGNC:24451 | |
| Strain, strain background (*Escherichia coli*) | BL21 Star (DE3) | Thermo Fisher | Invitrogen: C601003 | |
| Cell line (*H. sapiens*) | HEK-293T | ATCC | CRL-3216 | |
| Cell line (*H. sapiens*) | HEK-293T DCP1a KO | Elisa Izaurralde Lab | This paper | Cell line obtained from Elisa Izaurralde's laboratory. |
| Cell line (*H. sapiens*) | HEK-293T DCP1b KO | Elisa Izaurralde Lab | This paper | Cell line obtained from Elisa Izaurralde's laboratory. |
| Cell line (*H. sapiens*) | HEK-293T DCP1a/b KO | Elisa Izaurralde Lab | This paper | Cell line obtained from Elisa Izaurralde's laboratory. |
| Transfected construct (*H. sapiens*) | pT7-EGFP-C1-MBP | Elisa Izaurralde Lab | Addgene #146318 | |
| Transfected construct (*H. sapiens*) | pT7-EGFP-C1-HsDCP2 | Elisa Izaurralde Lab | Addgene #25031 | |
| Transfected construct (*H. sapiens*) | pT7-EGFP-C1-HsDCP2 E148Q | Elisa Izaurralde Lab | Addgene #147650 | |
| Transfected construct (*H. sapiens*) | pCIneo-V5-SBP-HsDCP2 | Elisa Izaurralde Lab | This paper | Plasmid obtained from Elisa Izaurralde's laboratory. |
| Transfected construct (*H. sapiens*) | pT7-EGFP-C1-HsDCP1a | Elisa Izaurralde Lab | Addgene #25030 | |
| Transfected construct (*H. sapiens*) | pT7-EGFP-C1-HsDCP1a EVH1 | Elisa Izaurralde Lab | Addgene #147089 | |
| Transfected construct (*H. sapiens*) | pT7-EGFP-C1-HsDCP1a 1–168 | Elisa Izaurralde Lab | This paper | Plasmid obtained from Elisa Izaurralde's laboratory. |
| Transfected construct (*H. sapiens*) | pT7-EGFP-C1-HsDCP1a 1–254 | Elisa Izaurralde Lab | This paper | Plasmid obtained from Elisa Izaurralde's laboratory. |
| Transfected construct (*H. sapiens*) | pT7-EGFP-C1-HsDCP1a 1–538 | Elisa Izaurralde Lab | This paper | Plasmid obtained from Elisa Izaurralde's laboratory. |
| Transfected construct (*H. sapiens*) | pT7-EGFP-C1-HsDCP1a ΔEVH1 | Elisa Izaurralde Lab | Addgene #147090 | |
| Transfected construct (*H. sapiens*) | pT7-EGFP-C1-HsDCP1a ΔHLM | Elisa Izaurralde Lab | This paper | Plasmid obtained from Elisa Izaurralde's laboratory. |
| Transfected construct (*H. sapiens*) | pT7-EGFP-C1-HsDCP1b | Elisa Izaurralde Lab | Addgene #147022 | |
| Transfected construct (*H. sapiens*) | pEF-DEST51 HA-PNRC1 | Giovanni Tonon Lab | Addgene #123294 | |
| Transfected construct (*H. sapiens*) | pCIneo-V5-SBP-MBP-PNRC2 | Elisa Izaurralde Lab | Addgene #147761 | |
| Transfected construct (*H. sapiens*) | pcDNA3.1-β-globin-6xMS2bs | Elisa Izaurralde Lab | *Chang et al., 2014* | |
| Transfected construct (*H. sapiens*) | pcDNA3.1-β-globin-GAPDH | Elisa Izaurralde Lab | *Chang et al., 2014* | |
| Transfected construct (*H. sapiens*) | pcDNA3.1-MS2-HA | Elisa Izaurralde Lab | Addgene #147544 | |

*Appendix 1 Continued on next page*

*Appendix 1 Continued*

| Reagent type (species) or resource | Designation | Source or reference | Identifiers | Additional information |
|---|---|---|---|---|
| Transfected construct (*H. sapiens*) | pcDNA3.1-MS2-HA-HsSMG7 | Elisa Izaurralde Lab | Addgene #147556 | |
| Transfected construct (*H. sapiens*) | pcDNA3.1-MS2-HA-Strep-HsSMG7 | Elisa Izaurralde Lab | This paper | Plasmid obtained from Elisa Izaurralde's laboratory. |
| Transfected construct (*H. sapiens*) | pcDNA3.1-MS2-HA-HsTNRC6A SD | Elisa Izaurralde Lab | Addgene #147556 | |
| Transfected construct (*H. sapiens*) | pcDNA3.1-MS2-HA-HsNanos1 | Elisa Izaurralde Lab | Addgene #147931 | |
| Transfected construct (*H. sapiens*) | pSUPER.puro-PNRC1 | Elisa Izaurralde Lab | This paper | Target sequence: CAAAGTTTAGT GATCCACCTTT |
| Transfected construct (*H. sapiens*) | pSUPER.puro-PNRC2 | Elisa Izaurralde Lab | This paper | Target sequence: AGTTGGAATTCTAGCTTAT |
| Antibody | Anti-V5 (mouse monoclonal) | Bio-Rad | Bio-Rad #MCA1360GA | WB (1:1000) |
| Antibody | Anti-GFP (rabbit polyclonal) | Elisa Izaurralde Lab | | For IP |
| Antibody | Anti-GFP (mouse monoclonal) | Roche | Roche #11814460001 | WB (1:2000) |
| Antibody | Anti-HA-HRP (mouse monoclonal) | Roche | Roche #12013819001 | WB (1:5000) |
| Antibody | Anti-Tubulin (mouse monoclonal) | Sigma-Aldrich | Sigma-Aldrich #T6199 | WB(1:5000) |
| Antibody | Anti-DCP2 (rabbit polyclonal) | Bethyl | A302-597A | WB (1:1000); IF (1:1000) |
| Antibody | Anti-DDX6 (rabbit polyclonal) | Bethyl | Bethyl #A300-461Z | WB (1:5000); IF (1:1000) |
| Antibody | Anti-EDC4 (mouse monoclonal) | Santa Cruz Biotechnology | sc-8418 | WB (1:1000); IF (1:1000) |
| Antibody | Anti-DCP1a (C-terminal region; Epitope: a.a 512–528) (rabbit polyclonal) | Sigma | D5444 | WB (1:1000) |
| Antibody | Anti-DCP1a (N-terminal region; Epitope: a.a 1–50) (Rabbit polyclonal) | aviva | ARP39353_T100 | WB (1:1000) |
| Antibody | Anti-DCP1b (Epitope: a.a 132–236) (Rabbit polyclonal) | Novus | NBP1-82018 | WB (1:1000) |
| Antibody | Anti-EDC3 (mouse monoclonal) | Abcam | Ab57780 | WB (1:1000) |
| Antibody | Anti-PatL1 (rabbit polyclonal) | Bethyl | A303-482A-M | WB (1:1000) |
| Antibody | Anti-4ET (rabbit polyclonal) | Bethyl | A300-706A | WB (1:1000); IF (1:1000) |
| Sequence-based reagent | β-globin_F | This paper | qPCR primers | ATGGTGCACCTG ACTCCTGAG |
| Sequence-based reagent | β-globin_R | This paper | qPCR primers | TTAGTGATACTTGT GGGCCAGGGC |
| Sequence-based reagent | GAPDH_F | *Weber and Chang, 2024* | qPCR primers | ctctgctcctcctgttcgacag |
| Sequence-based reagent | GAPDH_R | *Weber and Chang, 2024* | qPCR primers | ttcccgttctcagccttgacgg |
| Software, algorithm | RSEM | *Li and Dewey, 2011* | | |
| Other | GENCODE hg38 | *Frankish et al., 2019* | | |

*Appendix 1 Continued on next page*

*Appendix 1 Continued*

| Reagent type (species) or resource | Designation | Source or reference | Identifiers | Additional information |
|---|---|---|---|---|
| Software, algorithm | DESeq2 | *Love et al., 2014* | | |
| Software, algorithm | GSEA | *Subramanian et al., 2005* | | |
| Software, algorithm | clusterProfiler | *Wu et al., 2021a* | | |
| Software, algorithm | ComplexHeatmap | *Gu et al., 2016* | | |
| Software, algorithm | DoSurvive | *Wu et al., 2023* | | |

