## [Editor Report · eLife Assessment]

This study attempts to understand the functional roles of the human DCP1 paralogs in regulating RNA decay by DCP2. Using a combination of cellular-based assays and in vitro assays, the authors conclude that DCP1a/b plays a role in regulating DCP2 activity. While this revised version presents some new and interesting observations on human DCP1, the underlying data to support its claims remain **incomplete**. Overall, these results will be **useful** to the RNA community.

---

## [Referee Report · Reviewer #1 (Public review)]

Summary & Assessment:

The catalytic core of the eukaryotic decapping complex consists of the decapping enzyme DCP2 and its key activator DCP1. In humans, there are two paralogs of DCP1, DCP1a and DCP1b, that are known to interact with DCP2 and recruit additional cofactors or coactivators to the decapping complex; however, the mechanisms by which DCP1 activates decapping and the specific roles of DCP1a versus DCP1b, remain poorly defined. In this manuscript, the authors used CRISPR/Cas9-generated DCP1a/b knockout cells to begin to unravel some of the differential roles for human DCP1a and DCP1b in mRNA decapping, gene regulation, and cellular metabolism. While this manuscript presents some new and interesting observations on human DCP1 (e.g. human DCP1a/b KO cells are viable and can be used to investigate DCP1 function; only the EVH1 domain, and not its disordered C-terminal region which recruits many decapping cofactors, is apparently required for efficient decapping in cells; DCP1a and b target different subsets of mRNAs for decay and may regulate different aspects of metabolism), there is one key claim about the role of DCP1 in regulating DCP2-mediated decapping that is still incompletely or inconsistently supported by the presented data in this revised version of the manuscript.

Strengths & well-supported claims:

• Through in vivo tethering assays in CRISPR/Cas9-generated DCP1a/b knockout cells, the authors show that DCP1 depletion leads to significant defects in decapping and the accumulation of capped, deadenylated mRNA decay intermediates.

• DCP1 truncation experiments reveal that only the EVH1 domain of DCP1 is necessary to rescue decapping defects in DCP1a/b KO cells.

• RNA and protein immunoprecipitation experiments suggest that DCP1 acts as a scaffold to help recruit multiple decapping cofactors to the decapping complex (e.g. EDC3, DDX6, PATL1 PNRC1, and PNRC2), but that none of these cofactors are essential for DCP2-mediated decapping in cells.

• The authors investigated the differential roles of DCP1a and DCP1b in gene regulation through transcriptomic and metabolomic analysis and found that these DCP1 paralogs target different mRNA transcripts for decapping and have different roles in cellular metabolism and their apparent links to human cancers. (Although I will note that I can't comment on the experimental details and/or rigor of the transcriptomic and metabolomic analyses, as these are outside my expertise.)

Weaknesses & incompletely supported claims:

(1) One of the key mechanistic claims of the paper is that "DCP1a can regulate DCP2's cellular decapping activity by enhancing DCP2's affinity to RNA, in addition to bridging the interactions of DCP2 with other decapping factors. This represents a pivotal molecular mechanism by which DCP1a exerts its regulatory control over the mRNA decapping process." Similar versions of this claim are repeated in the abstract and discussion sections. However, this claim appears to be at odds with the observations that: (a) in vitro decapping assays with immunoprecipitated DCP2 show that DCP1 knockout does not significantly affect the enzymatic activity of DCP2 (Fig 2C&D; I note that there may be a very small change in DCP2 activity shown in panel D, but this may be due to slightly different amounts of immunoprecipitated DCP2 used in the assay); and (b) the authors show only weak changes in relative RNA levels immunoprecipitated by DCP2 with versus without DCP1 (~2-3 fold change in Fig 3H, where expression of the EVH1 domain, previously shown in this manuscript to fully rescue the DCP1 KO decapping defects in cells, looks to be almost within error of the control in terms of increasing RNA binding). If DCP1 pivotally regulates decapping activity by enhancing RNA binding to DCP2, why is no difference in in vitro decapping activity observed in the absence of DCP1, and very little change observed in the amounts of RNA immunoprecipitated by DCP2 with the addition of the DCP1 EVH1 domain?

In the revised manuscript and in their response to initial reviews, the authors rightly point out that in vivo effects may not always be fully reflected by or recapitulated in in vitro experiments due to the lack of cellular cofactors and simpler environment for the in vitro experiment, as compared to the complex environment in the cell. I fully agree with this of course! And further completely agree with the authors that this highlights the critical importance of in cell experiments to investigate biological functions and mechanisms! However, because the in vitro kinetic and IP/binding data both suggest that the DCP1 EVH1 domain has minimal to no effects on RNA decapping or binding affinity, while the in cell data suggest the EVH1 domain alone is sufficient to rescue large decapping defects in DCP1a/b KO cells (and that all the decapping cofactors tested were dispensable for this), I would argue there is insufficient evidence here to make a claim that (maybe weakly) enhanced RNA binding induced by DCP1 is what is regulating the cellular decapping activity. Maybe there are as-yet-untested cellular cofactors that bind to the EVH1 domain of DCP1 that change either RNA recruitment or the kinetics of RNA decapping in cells; we can't really tell from the presented data so far. Furthermore, even if it is the case that the EVH1 domain modestly enhances RNA binding to DCP2, the authors haven't shown that this effect is what actually regulates the large change in DCP2 activity upon DCP1 KO observed in the cell.

Overall, while I absolutely appreciate that there are many possible reasons for the differences observed in the in vitro versus in cell RNA decapping and binding assays, because this discrepancy between those data exists, it seems difficult to draw any clear conclusions about the actual mechanisms by which DCP1 helps regulate RNA decapping by DCP2. For example, in the cell it could be that DCP1 enhances RNA binding, or recruits unidentified cofactors that themselves enhance RNA binding, or that DCP1 allosterically enhances DCP2-mediated decapping kinetics, or a combination of these, etc; my point is that without in vitro data that clearly support one of those mechanisms and links this mechanism back to cellular DCP2 decapping activity (for example, in cell data that show EVH1 mutants that impair RNA binding fail to rescue DCP1 KO decapping defects), it's difficult to attribute the observed in cell effects of DCP1a/b KO and rescue by the EVH1 domain directly to enhancement of RNA binding (precisely because, as the authors describe, the decapping process and regulation may be very complex in the cell!).

This contradiction between the in vitro and in-cell decapping data undercuts one of the main mechanistic takeaways from the first half of the paper; I still think this conclusion is overstated in the revised manuscript.

Additional minor comment:

• Related to point (1) above, the kinetic analysis presented in Fig 2C shows that the large majority of transcript is mostly decapped at the first 5 minute timepoint; it may be that DCP2-mediated decapping activity is actually different in vitro with or without DCP1, but that this is being missed because the reaction is basically done in less than 5 minutes under the conditions being assayed (i.e. these are basically endpoint assays under these conditions). It may be that if kinetics were done under conditions to slow down the reaction somewhat (e.g. lower Dcp2 concentration, lower temperatures), so that more of the kinetic behavior is captured, the apparent discrepancy between in vitro and in-cell data would be much less. Indeed, previous studies have shown that in yeast, Dcp1 strongly activates the catalytic step (kcat) of decapping by ~10-fold, and reduces the KM by only ~2 fold (Floor et al, NSMB 2010). It might be beneficial to use purified proteins here, if possible, to better control reaction conditions.

In their response to initial reviews, the authors comment that they tried to purify human DCP2 from E coli, but were unable to obtain active enzyme in this way. Fair enough! I will only comment that just varying the relative concentration of immunoprecipitated DCP2 would likely be enough to slow down the reaction and see if activity differences are seen in different kinetic regimes, without the need to obtain fully purified / recombinant Dcp2.

---

## [Author Response]

The following is the authors’ response to the original reviews.

**Public Reviews:**

**Reviewer #1 (Public Review):**
Weaknesses & incompletely supported claims:(1) A central mechanistic claim of the paper is that "DCP1a can regulate DCP2's cellular decapping activity by enhancing DCP2's affinity to RNA, in addition to bridging the interactions of DCP2 with other decapping factors. This represents a pivotal molecular mechanism by which DCP1a exerts its regulatory control over the mRNA decapping process." Similar versions of this claim are repeated in the abstract and discussion sections. However, this appears to be entirely at odds with the observation from in vitro decapping assays with immunoprecipitated DCP2 that showed DCP1 knockout does not significantly affect the enzymatic activity of DCP2 (Figures 2B-D; I note that there may be a very small change in DCP2 activity shown in panel C, but this may be due to slightly different amounts of immunoprecipitated DCP2 used in the assay, as suggested by panel D). If DCP1 pivotally regulates decapping activity by enhancing RNA binding to DCP2, why is no difference in decapping activity observed in the absence of DCP1?Furthermore, the authors show only weak changes in relative RNA levels immunoprecipitated by DCP2 with versus without DCP1 (~2-3 fold change; consistent with the Valkov 2016 NSMB paper, which shows what looks like only modest changes in RNA binding affinity for yeast Dcp2 +/- Dcp1). Is the argument that only a 2-3 fold change in RNA binding affinity is responsible for the sizable decapping defects and significant accumulation of deadenylated intermediates observed in cells upon Dcp1 depletion? (and if so, why is this the case for in-cell data, but not the immunoprecipitated in vitro data?)

We appreciate the reviewer's thoughtful comments on our paper. The reviewer points out an apparent contradiction between the claim that DCP1a regulates DCP2's cellular decapping activity and the observation that knocking out DCP1a does not significantly affect DCP2's enzymatic activity in vitro. However, it is important to underscore the challenge of reconciling differences between in vitro and in vivo experiments in scientific research. Although in vitro systems provide a controlled environment, they have inherent limitations that often fail to capture the complexities of cellular processes. Our in vitro experiments used immunoprecipitated proteins to ensure the presence of relevant factors, but these experiments cannot fully replicate the precise stoichiometry and dynamic interactions present in a cellular environment. Furthermore, the limited volume in vitro can actually facilitate reactions that may not occur as readily in the complex and heterogeneous environment of a cell. Therefore, the lack of a significant difference in decapping activity observed in vitro does not necessarily negate the regulatory role of DCP1 in the cellular context. Rather, it underscores our previous oversight of DCP1's importance in the decapping process under in vitro conditions. The conclusions regarding DCP1's regulatory mechanisms remain valid and supported by the presented evidence, especially when considering the inherent differences between in vitro and in vivo experimental conditions. It is precisely because of these differences that we recognized our previous underestimation of DCP1's significance. Therefore, our subsequent experiments focused on elucidating DCP1's regulatory mechanisms in the decapping process

The authors acknowledge this apparent discrepancy between the in vitro DCP2 decapping assays and in-cell decapping data, writing: "this observation could be attributed to the inherent constraints of in vitro assays, which often fall short of faithfully replicating the complexity of the cellular environment where multiple factors and cofactors are at play. To determine the underlying cause, we postulated that the observed cellular decapping defect in DCP1a/b knockout cells might be attributed to DCP1 functioning as a scaffold." This is fair. They next show that DCP1 acts as a scaffold to recruit multiple factors to DCP2 in cells (EDC3, DDX6, PatL1, and PNRC1 and 2). However, while DCP1 is shown to recruit multiple cofactors to DCP2 (consistent with other studies in the decapping field, and primarily through motifs in the Dcp1 C-terminal tail), the authors ultimately show that *none* of these cofactors are actually essential for DCP2-mediated decapping in cells (Figures 3A-F). More specifically, the authors showed that the EVH1 domain was sufficient to rescue decapping defects in DCP1a/b knockout cells, that PNRC1 and PNRC2 were the only cofactors that interact with the EVH1 domain, and finally that shRNA-mediated PNRC1 or PNCR2 knockdown has no effect on in-cell decapping (Figures 3E and F). Therefore, based on the presented data, while DCP1 certainly does act as a scaffold, it doesn't seem to be the case that the major cellular decapping defect observed in DCP1a/b knockout is due to DCP1's ability to recruit specific cofactors to DCP2.

The findings that none of the decapping cofactors recruited by DCP1 to DCP2 are essential for decapping in cells further underscore the complexity of the decapping process in vivo. This observation suggests that while DCP1's scaffolding function is crucial for recruiting cofactors, the decapping process likely involves additional layers of regulation that are not fully captured by our current understanding of DCP1. Furthermore, the reviewer mentions that the observed changes in RNA binding affinity (approximately 2-3 fold) in our in vitro experiments seem relatively modest. While these changes may appear insignificant in vitro, their cumulative impact in the dynamic cellular environment could be substantial. Even minor perturbations in RNA binding affinity can trigger cascading effects, leading to significant changes in decapping activity and the accumulation of deadenylated intermediates upon Dcp1 depletion. Cellular processes involve complex networks of interrelated events, and small molecular changes can result in amplified biological outcomes. The subtle molecular variations observed in vitro may translate into significant phenotypic outcomes within the complex cellular environment, underscoring the importance of DCP1a's regulatory role in the cellular decapping process.

So as far as I can tell, the discrepancy between the in vitro (DCP1 not required) and in-cell (DCP1 required) decapping data, remains entirely unresolved. Therefore, I don't think that the conclusions that DCP1 regulates decapping by (a) changing RNA binding affinity (authors show this doesn't matter in vitro, and that the change in RNA binding affinity is very small) or (b) by bridging interactions of cofactors with DCP2 (authors show all tested cofactors are dispensable for robust in-cell decapping activity), are supported by the evidence presented in the paper (or convincingly supported by previous structural and functional studies of the decapping complex).

We have addressed the reconciliation of differences between in vitro and in vivo experiments in the revised manuscript and emphasized the importance of considering cellular interactions when interpreting our findings.

(2) Related to the RNA binding claims mentioned above, are the differences shown in Figure 3H statistically significant? Why are there no error bars shown for the MBP control? (I understand this was normalized to 1, but presumably, there were 3 biological replicates here that have some spread of values?). The individual data points for each replicate should be displayed for each bar so that readers can better assess the spread of data and the significance of the observed differences. I've listed these points as major because of the key mechanistic claim that DCP1 enhances RNA binding to DCP2 hinges in large part on this data.

Thank you for your feedback. Regarding your comments on the statistical significance of the differences shown in Figure 3H and the absence of error bars for the MBP control, we will address these concerns in the revised manuscript. We’ll include individual data points for the three biological replicates and corresponding statistical analysis to more clearly demonstrate the data spread and significance of the observed differences.

(3) Also related to point (1) above, the kinetic analysis presented in Figure 2C shows that the large majority of transcript is mostly decapped at the first 5-minute timepoint; it may be that DCP2-mediated decapping activity is actually different in vitro with or without DCP1, but that this is being missed because the reaction is basically done in less than 5 minutes under the conditions being assayed (i.e. these are basically endpoint assays under these conditions). It may be that if kinetics were done under conditions to slow down the reaction somewhat (e.g. lower Dcp2 concentration, lower temperatures), so that more of the kinetic behavior is captured, the apparent discrepancy between in vitro and in-cell data would be much less. Indeed, previous studies have shown that in yeast, Dcp1 strongly activates the catalytic step (kcat) of decapping by ~10-fold, and reduces the KM by only ~2 fold (Floor et al, NSMB 2010). It might be beneficial to use purified proteins here (only a Western blot is used in Figure 2D to show the presence of DCP2 and/or DCP1, but do these complexes have other, and different, components immunoprecipitated along with them?), if possible, to better control reaction conditions.This contradiction between the in vitro and in-cell decapping data undercuts one of the main mechanistic takeaways from the first half of the paper. This needs to be addressed/resolved with further experiments to better define the role of DCP1-mediated activation, or the mechanistic conclusions significantly changed or removed.

We genuinely appreciate the reviewer’s insightful comments on the kinetic analysis presented in Figure 2C. Your astute observation regarding the potential influence of reaction duration on the interpretation of in vitro decapping activity, especially in the absence of DCP1, is well-received. The time-sensitive nature of our experiments, as you rightly pointed out, might not fully capture the nuanced kinetic behaviors. In addition, the DCP2 complex purified from cells could not be precisely quantified. In response to your suggestion, we attempted to purify human DCP2 protein from *E. coli*; however, regrettably, the purified protein failed to exhibit any enzymatic activity. This disparity may be attributed to species differences.

Considering the reviewer’s valuable insights, our revised manuscript emphasized that purified DCP2 from cells exhibits activity regardless of the presence of DCP1. This adjustment aims to provide a clearer perspective on our findings and to better align with the nuances of our experimental design and the meticulous consideration of the results.

(4) The second half of the paper compares the transcriptomic and metabolic profiles of DCP1a versus DCP1b knockouts to reveal that these target a different subset of mRNAs for degradation and have different levels of cellular metabolites. This is a great application of the DCP1a/b KO cells developed in this paper and provides new information about DCP1a vs b function in metazoans, which to my knowledge has not really been explored at all. However, the analysis of DCP1 function/expression levels in human cancer seems superficial and inconclusive: for example, the authors conclude that "...these findings indicate that DCP1a and DCP1b likely have distinct and non-redundant roles in the development and progression of cancer", but what is the evidence for this? I see that DCP1a and b levels vary in different cancer cell types, but is there any evidence that these changes are actually linked to cancer development, progression, or tumorigenesis? If not, these broader conclusions should be removed.

Thank you to the reviewer for pointing out that such a description may be misleading. We have removed our previous broader conclusion and revised our sentences. To further explore the potential impact of DCP1a and DCP1b on cancer progression, we examined the association between the expression levels of DCP1a and DCP1b and progression-free interval (PFI). We have incorporated this information into our revised manuscript.

(5) The authors used CRISPR-Cas9 to introduce frameshift mutations that result in premature termination codons in DCP1a/b knockout cells (verified by Sanger sequencing). They then use Western blotting with DCP1a or DCP1b antibodies to confirm the absence of DCP1 in the knockout cell lines. However, the DCP1a antibody used in this study (Sigma D5444) is targeted to the C-terminal end of DCP1a. Can the authors conclusively rule out that the CRISPR/Cas-generated mutations do not result in the production of truncated DCP1a that is just unable to be detected by the C-terminally targeted antibody? While it is likely the introduced premature termination codon in the DCP1a gene results in nonsense-mediated decay of the resulting transcript, this outcome is indeed supported by the knockout results showing large defects in cellular decapping which can be rescued by the addition of the EVH1 domain, it would be better to carefully validate the success of the DCP1a knockout and conclusively show no truncated DCP1a is produced by using N-terminally targeted DCP1a antibodies (as was the case for DCP1b).

Thank you for your insightful comment regarding the validation of our DCP1a/b knockout cell line. We acknowledge your point about the DCP1a C-terminal targeting of the Sigma D5444 antibody used in our Western blot analysis. We agree that we cannot definitively rule out the possibility of truncated DCP1a protein production solely based on the lack of full-length protein detection. To address this limitation, we utilized a commercial information available N-terminally targeted DCP1a antibody (aviva ARP39353_T100) in a Western blot analysis. This will allow us to comprehensively detect any truncated protein fragments remaining after the CRISPR-Cas9-generated frameshift mutation.

Some additional minor comments:• More information would be helpful on the choice of DCP1 truncation boundaries; why was 1-254 chosen as one of the truncations?

Thank you for the reviewer's comment and suggestion. Regarding the choice of DCP1 1-254 truncation boundaries based on the predicted structure from AlphaFoldDB (A0A087WT55). We will include this information in the revised manuscript.

• Figure S2D is a pretty important experiment because it suggests that the observed deadenylated intermediates are in fact still capped; can a positive control be added to these experiments to show that removal of cap results in rapid terminator-mediated degradation?

Unfortunately, due to our institution's current laboratory safety policies, we are unable to perform experiments involving the use of radioactive isotopes such as 32P. Therefore, while adding the suggested positive control experiment to demonstrate rapid RNA degradation upon decapping would further validate our interpretation, we regret that we cannot carry out this experiment at the moment. However, the observed deadenylated intermediates in Figure S2D match the predicted size of capped RNA fragments, and not the expected sizes of degradation products after decapping. Furthermore, previous literature has well-established that for these types of RNAs, decapping leads directly to rapid 5' to 3' exonuclease-mediated degradation, without producing stable deadenylated intermediates. Thus, we believe that the current data is sufficient to support our conclusion that the deadenylated intermediates retain the 5' cap structure.

**Reviewer #2 (Public Review):**
Weaknesses:The direct targets of DCP1a and/or DCP1b were not determined as the analysis was restricted to RNA-seq to assess RNA abundance, which can be a result of direct or indirect regulation by DCP1a/b.

Thank you for raising this important point. In our study, we acknowledge that the use of RNA-seq to assess RNA abundance provides a broad overview of the regulatory impacts of DCP1a and DCP1b. This method captures changes in RNA levels that may arise from both direct and indirect regulatory actions of these proteins. While we did not directly determine the targets of DCP1a and DCP1b, the data obtained from our RNA-seq analysis serve as a foundational step for future targeted experiments, which could include techniques such as RIP-seq, to delineate the direct targets of DCP1a and DCP1b more precisely. We believe that our current findings contribute valuable information to the field and pave the way for these subsequent analyses.

P-bodies appear to be larger in human cells lacking DCP1a and DCP1b but a lack of image quantification prevents this conclusion from being drawn.

Thank you for the reviewer’s valuable feedback. We have addressed the reviewer’s concern regarding P-bodies' size in human cells lacking DCP1a and DCP1b. We have now performed image quantification and can confirm that P-bodies are indeed larger in these cells.

The lack of details in the methodology and figure legends limit reader understanding.

We acknowledge the reviewer's concerns regarding the level of detail provided in the methodology and figure legends. To address this, we are committed to enhancing both sections with additional details and clarifications in our revised manuscript. Thank you for bringing this to our attention.

**Recommendations for the authors:**

**Reviewer #1 (Recommendations For The Authors):**
(1) To me, the second half of the paper comparing DCP1a and DCP1b is in many ways distinct from the first half and could stand on its own as an interesting paper if this comparative analysis is explored a little deeper (maybe by validating some of the differences in decay observed for individual mRNAs targeted by DCP1a versus DCP1b, by measuring and comparing the decay rates of some individual transcripts under differential control by DCP1a vs b?), and revising the conclusions about links to cancer as mentioned above. I think these later comparative results in the paper present the most new and interesting data concerning DCP1 function in humans (especially since I think the mechanistic conclusions from the first half aren't well supported yet or are at least inconsistent), but when I read these later sections of the paper I struggle to understand the key takeaways from the transcriptomic and metabolomic data.

Thank you for the reviewer's suggestions. Estimating the decay rates of individual transcripts within the transcriptomes of DCP1a_KO, DCP1b_KO, and wild type can provide insight into the direct targets of DCP1a or DCP1b. However, this requires either time-series RNA-seq or specialized sequencing technologies such as Precision Run-On sequencing (PRO-seq) or RNA Approach to Equilibrium Sequencing (RATE-Seq). Unfortunately, we lack the necessary dataset in our project to estimate the decay rates for the potential targets identified in our RNA-seq data. Despite this limitation, we acknowledge the potential of this approach in identifying the true targets of DCP1a and DCP1b and have included this idea in our discussion.

(2) I think it would be helpful to add a little more descriptive or narrative language to the figure legends (I know some of them are already quite long!) so that readers can follow the general idea of the experiment through the figure legend as well as the main text; as written, the figure legends are mostly exclusively technical details, so it can be hard to parse what experiment is being carried out in some cases.

Thank you for the reviewer’s suggestion, we will strive to improve the language of the figure legends to include technical details while clearly conveying the main idea of the experiment. We will ensure that the language of the figure legends is more readable and comprehensible so that readers can more easily parse what experiment is being carried out.

**Reviewer #2 (Recommendations For The Authors):**
Suggestions for improved or additional experiments, data, or analyses:The use of RNA-seq to measure RNA abundance in DCP1a and/or b knockout cells can give some insight into both the indirect and direct effects of DCP1a/b on gene expression but cannot identify the direct targets of these genes. Rather, global analysis of RNA stability or capturing uncapped RNA decay intermediates would allow the authors to conclude they have identified direct targets of DCP1a and/or b. Without such analyses, the interpretation of these data should be scaled back to clearly state that RNA levels can be altered through indirect effects of DCP1a/b absence throughout the text.

We appreciate the reviewer's suggestion. We have modified our sentences to emphasize that the dysregulated genes could be caused by both direct and indirect effects.

A control/randomly generated gene list should be analyzed for GO terms to determine whether the enrichment of cancer-related pathways in the differentially expressed genes in the DCP1a/b knockout cells is meaningful.

Thank you for the reviewer's comment. We shuffled our gene list and reperformed the pathway enrichment analysis in Figure 4C and 4D 1,000 times. We focused on the following cancer-related pathways: E2F targets, MTORC1 signaling, G2M checkpoint, MYC target V1, EMT transition, KRAS signaling DN, P53 pathway, and NOTCH signaling pathways. We then calculated how many times the q-values obtained from the shuffled gene list were more significant than the q-value obtained from our real data. In four of the eight pathways (E2F targets, MTORC1 signaling, G2M checkpoint, and MYC target v1), none of the shuffled gene lists resulted in a q-value smaller than the real one. In the other four pathways (EMT transition, KRAS signaling DN, P53 pathway, and NOTCH signaling pathways), the q-values were smaller than the real q-value 2, 11, 4, and 4 times out of the 1000 shuffles. Based on the shuffled results, we conclude that the transcriptome of DCP1a/b knockout cells is statistically enriched in these cancer-related pathways.

**Author response image 1. sa2fig1:** Distribution of q-values resulting from the Gene Set Enrichment Analysis (GSEA) conducted on 1,000 shuffled gene lists for eight cancer-related pathways. The q-values derived from Figure 4C and 4D are indicated by red (DCP1a_KO) and blue (DCP1b_KO) dashed lines, respectively. Some q-values derived from Figure 4C are too small to be labeled on the plots, such as in E2F targets (q value: 5.87E-07), MTORC1 signaling (q values: 6.59E-07 and 1.58E-06 for DCP1a_KO and DCP1b_KO, respectively), MYC target V1 (q value: 0.004644174 for DCP1a_KO), etc. The numbers x/1000 indicate how often the shuffled q-values were smaller than the real q-value out of 1,000 permutations.

Comparisons of the DCP1a and/or b knockout RNA-seq results should be done to published datasets such as those published by Luo et al., Cell Chemical Biology (2021) to determine whether there are common targets with DCP2 and validate the reported findings.

Thank you for reviewer’s suggestion. We compared the upregulated genes from DCP1a_KO, DCP1b_KO, and DCP1a/b_KO cell lines with the 91 targets of DPC2 identified by Luo et al. in Cell Chemical Biology (2021). Only EPPK1 was found to be overlapped between the potential DCP1b_KO targets and the targets of DCP2. No genes were found to be overlapped between the potential DCP1a_KO targets and the targets of DCP2. However, three genes, TES, PAX6, and C18orf21, were found to be overlapped between the significantly upregulated DEGs of DCP1a/b_KO and the targets of DCP2. We have included this information in the discussion section.

The RNA tethering assays are not clear and are difficult to interpret without further controls to delineate the polyadenylated and deadenylated species.

Thank you for the reviewer’s feedback. We acknowledge that the reviewer might harbor some doubts regarding the outcomes of the RNA tethering assays. Nonetheless, this methodology is well-established and has also found extensive application across many studies. We are committed to enhancing the clarity of our experiment’s details and results within the figure legends and textual descriptions.

The representative images of p-bodies clearly show that DCP1a/b KO cells have larger p-bodies than the wild-type cells. The authors should quantify p-body size in each image set as the current interpretation of the data is that there is no difference in size or number of p-bodies, but the data suggest otherwise.

Thank you very much for the reviewer’s insightful comments and for drawing our attention to the need to quantify p-body sizes in DCP1a/b KO and wild-type cells. We agree with the reviewer’s assessment that the representative images suggest a difference in p-body size between DCP1a/b KO cells and wild-type cells, which we initially overlooked. We will revise our manuscript accordingly to include these findings, ensuring that our interpretation of the data aligns with the observed differences.

Statistical analysis of the Figure 2C results should be included because the difference between the wild-type and Dco1a/b KO cells with GFP-DCP2 looks significantly different but is interpreted in the text as not significant.

Thank you for pointing out the need for a statistical analysis of the results shown in Figure 2C. We acknowledge that the visual difference between the wild-type and Dco1a/b KO cells with GFP-DCP2 suggests a significant variation, which may not have been clearly communicated in our text. We will conduct the necessary statistical analysis to substantiate the observations made in Figure 2C. Furthermore, we would like to emphasize that our primary focus was to demonstrate that purified DCP2 within cells retains its activity even in the absence of DCP1. This critical point will be highlighted and clarified in the revised version of our manuscript to prevent any misunderstanding.

Recommendations for improving the writing and presentation:Additional context including what is known about the role of dcp1 in decapping from the decades of work in yeast and other model organisms should be incorporated into the introduction and discussion sections.

Thank you for the reviewer’s suggestion. We will incorporate additional context about the function and significance of DCP1 in decapping processes within our revised manuscript's introduction and discussion sections.

Details should be provided within the figure legends and methods section on experimental approaches and the number of replicates and statistical analyses used throughout the manuscript. For example, it is not clear whether western blots or RNA-IP experiments were performed more than once as representative images are shown.

Thank you for the reviewer’s suggestion. In the figure legends and methods section, we will provide more details about the experimental methods, number of replicates, and statistical analyses. Regarding the Western blots and RNA-IP experiments the reviewer mentioned, we performed multiple experiments and presented representative images in the manuscript. We will clarify this in the revised manuscript to eliminate potential confusion.

The rationale for performing metabolic profiling is not clear.

We appreciate the reviewer's thoughtful feedback. The rationale behind conducting metabolic profiling in our study is rooted in its efficacy as a valuable tool for deciphering the consequences of specific gene mutations, particularly those closely associated with phenotypic changes or final metabolic pathways. Our objective is to utilize metabolic profiling to unravel the distinct biofunctions of DCP1a and DCP1b. By employing this approach, we aim to gain insights into the intricate metabolic alterations that result from the absence of these genes, thereby enhancing our understanding of their roles in cellular processes. We recognize the necessity of clearly presenting this rationale and promise to bolster the articulation of these points in the revised version of our manuscript to ensure the clarity and transparency of our research motivation.

Details in the methods section should be included for the CRISPR/Cas9-mediated gene editing validation. The Sangar sequencing results presented in Figure S1b should be explained. The entire western blot(s) should be shown in Figure S1A to give confidence the Dcp1a/b KO cells are not expressing truncated proteins and the epitopes of the antibodies used to detect Dcp1a/b should be described. The northern blot probes should be described and sequences included. The transcriptomics method should be detailed.

Thank you for your feedback, in the revised manuscript we will detail the CRISPR/Cas9 gene editing validation, explain the Sanger sequencing results in Figure S1b, show the full Western blot in Figure S1A to confirm that the Dcp1a/b knockout cells are not expressing truncated proteins, describe the Northern blot probes used, and detail the transcriptomics method, all to ensure clarity and comprehensiveness in our experimental procedures and results.

A diagram showing the RNA tethering assays with labels corresponding to all blots/gels should be provided.

Thank you for your suggestion. We will provide a diagram showing the RNA tethering assays with labels corresponding to all blots/gels in our revised manuscript. This will help readers better understand our experimental design and results.

The statement, "This suggests that the disruption of the decapping process in DCP1a/b-knockout cells results in the accumulation of unprocessed mRNA intermediates" regarding the results of the RNA-seq assay is not supported by the evidence as RNA-seq does not measure RNA decay intermediates or RNA decay rates.

Thank you for the reviewer’s comment. We agree with that RNA-seq experiments indeed do not directly measure RNA decay intermediates or RNA decay rates. Our statement could have caused confusion, and we have therefore removed this sentence from the manuscript.

Minor corrections to the text and figures:Figure S6A is uninterpretable as presented.

Thank you for the reviewer’s valuable feedback. We have taken note and made improvements. We have simplified Figure S6A to enhance its interpretability, hoping that the current version will make it easier for the readers to understand.